# TELL ME HABIBI, IS IT REAL OR FAKE? 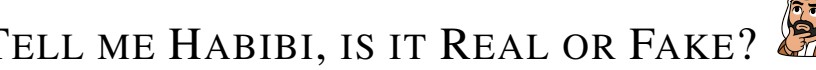

**Kartik Kuckreja,**[λ] **Parul Gupta,**[ξ] **Injy Hamed,**[λ]
**Thamar Solorio,**[λ] **Muhammad Haris Khan,**[λ] **Abhinav Dhall**[ξ]
[λ]MBZUAI    [ξ]Monash University *

## ABSTRACT

Deepfake generation methods are evolving fast, making fake media harder to detect and raising serious societal concerns. Most deepfake detection and dataset creation research focuses on monolingual content, often overlooking the challenges of multilingual and code-switched speech, where multiple languages are mixed within the same discourse. Code-switching, especially between Arabic and English, is common in the Arab world and is widely used in digital communication. This linguistic mixing poses extra challenges for deepfake detection, as it can confuse models trained mostly on monolingual data. To address this, we introduce **ArEnAV**, the first large-scale Arabic-English audio-visual deepfake dataset featuring intra-utterance code-switching, dialectal variation, and monolingual Arabic content. It **contains 387k videos and over 765 hours of real and fake videos**. Our dataset is generated using a novel pipeline integrating four Text-To-Speech and two lip-sync models, enabling comprehensive analysis of multilingual multimodal deepfake detection. We benchmark our dataset against existing monolingual and multilingual datasets, SOTA deepfake detection models, and a human evaluation, highlighting its potential to advance deepfake research. The dataset is public.

## 1 INTRODUCTION

Deepfake technologies, involving the artificial generation and manipulation of audio-visual content, have rapidly advanced, significantly complicating the task of distinguishing real media from synthetic creations. The potential misuse of deepfakes for misinformation, defamation, or impersonation presents profound societal risks, driving substantial research into their detection. Although initial deepfake research primarily focused on manipulating individual modalities, audio-only (Todisco et al., 2019) or video-only (Jiang et al., 2020; Kwon et al., 2021; Li et al., 2020b), recent developments increasingly consider joint manipulation of audio and visual streams for more realistic synthesis.

A significant gap remains in existing deepfake datasets (Table 1), which largely overlook multilingual scenarios, particularly code-switching (CSW), despite its global prevalence among bilingual speakers. In the Arab world, CSW is a prominent feature of daily communication, serving not only as a linguistic tool but also as a marker of cultural identity and social context (Hamed et al., 2025). Arabic speakers frequently alternate between Arabic and English within the same sentence, such as:

مهم جداً *deepfake detection* موضوع ال (The topic of *deepfake detection* is very important).

This challenge is compounded by the diglossic nature of Arabic (Ferguson, 1959), comprising of two main varieties: Modern Standard Arabic (MSA) and Dialectal Arabic (DA). MSA functions as a lingua franca across the Arab world and is primarily used in formal settings. DA, belonging to each country, is used in everyday conversations and informal writing. Given that Arabic is one of the most widely spoken languages worldwide, ranked fifth by number of Standard Arabic speakers (Ethnologue, 2025), handling its diglossic variation and code-switching phenomena is essential for building deepfake detection systems that address the linguistic diversity in real-world media content.

Recent studies provide compelling evidence of how common CSW is among Arabic speakers. The ZAEBUC-Spoken corpus (Hamed et al., 2024) reveals that approximately 19% of spoken utterances

*{kartik.kuckreja,injy.hamed,thamar.solorio,muhammad.haris}@mbzuai.ac.ae
{parul,abhinav.dhall}@monash.edu

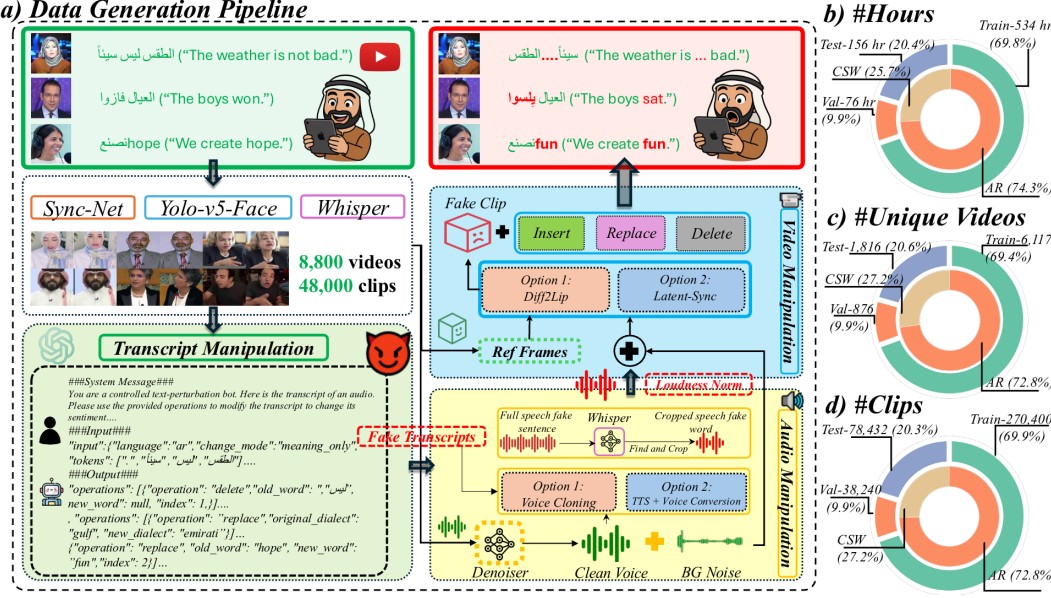

Figure 1: a) We show the data generation pipeline for ArEnAV dataset. In a) input videos are analysed for audio, face, and text extraction. Using few-shot prompts with GPT-4.1-mini, CSW-based spoken text manipulation is performed. This is followed by speech and face enactment generation. b-d) The plots show the data splits and CSW distribution. Here is an example of CSW input and manipulated text with translations in parentheses: نصنع hope ("We create hope.") –> نصنع fun ("We create fun.")

exhibit CSW, having an average of 44% English words. The corpus also highlights the presence of diglossic CSW between Arabic variants. Similarly, the ArzEn corpus (Hamed et al., 2020) demonstrates high CSW frequency, where 63% of utterances involve CSW with approximately 19% of words being English. These findings highlight the extent to which Arabic-English CSW is not merely an incidental phenomenon, but a widespread communicative strategy. Despite its ubiquity, deepfake detection systems remain largely ill-equipped to handle such language alternation, focusing predominantly on monolingual data. Addressing this important oversight, our work seeks to bridge this critical gap by introducing the first Arabic-English CSW audio-visual deepfake dataset, thus advancing the field toward more relevant detection systems. Our core contributions are as follows:

- Introduction of ArEnAV, the first large-scale Arabic-English audio-visual deepfake dataset featuring intra-utterance code-switching and dialectal variation, including both bilingual and diglossic switching across Modern Standard Arabic, Egyptian, Levantine, and Gulf dialects, addressing a critical gap in multilingual deepfake research.

- A novel data generation pipeline tailored to English and Arabic (MSA and dialect-rich content), integrating four TTS (Text to Speech) and two lip-sync models.

- A comprehensive analysis contrasting our dataset against existing monolingual and multilingual datasets, existing state-of-the-art (SOTA) deepfake detection models, and a detailed User Study, underscoring its unique difficulty in detection by models and humans alike.

## 2 RELATED WORK

Early deepfake research was predominantly monolingual and modality-specific. Initial significant contributions included video manipulation techniques such as FaceSwap and Face2Face as introduced by Thies et al. (2020), which led to seminal datasets like FaceForensics++ (Rössler et al., 2019) and the DeepFake Detection Challenge (DFDC) (Dolhansky et al., 2020a). These datasets primarily provided facial manipulations within single-language contexts, focusing largely on visual realism.

Parallel to video deepfake advancements, audio deepfakes evolved rapidly, driven by progress in text-to-speech (TTS) synthesis, voice conversion, and generative audio models such as Tacotron (Wang et al., 2017). Datasets like ASVspoof (Wang et al., 2020) and WaveFake (Frank & Schönherr,

2021) contributed significantly by providing benchmarks to evaluate audio manipulation detection methods, albeit still largely restricted to English.

In recent years, research has expanded towards joint audio-visual deepfake manipulations. Datasets such as FakeAVCeleb (Khalid et al., 2022) showcased realistic lip-synced speech synthesis in tandem with facial manipulations. AV-Deepfake1M (Cai et al., 2024a) further advanced this domain by automating transcript alterations to create nuanced, localized audio-visual deepfakes, highlighting the necessity of detecting temporally and contextually subtle manipulations.

Recently, there has been increased focus on multilingual audio deepfakes. These efforts have revealed key limitations in generalizing detection models across languages and proposed new resources to address these challenges. Marek et al. (2024) conducted a comprehensive study on cross-lingual deepfake detection, showing that models trained on one language often fail to generalize effectively to others, underscoring the role of language-specific phonetic and prosodic features in model performance. Multilingual audio-visual datasets emerged even more recently to address the global dimension of deepfake threats. The PolyGlotFake dataset (Hou et al., 2024) contains audio-visual deepfakes across seven languages. Although the dataset covers a wide range of language, the size of the real data is significantly small. Nonetheless, these multilingual datasets remain limited to either monolingual or monomodal scenarios within each single instance, ignoring the prevalent reality of intra-utterance language switching.

Our work directly responds to this critical gap. Unlike previous studies, we not only incorporate multilingual content but also explicitly generate intra-utterance code-switched audio-visual deepfakes. We leverage SOTA TTS and lip sync methodologies adapted for multilingual use, resulting in realistic, diverse, and challenging benchmarks.

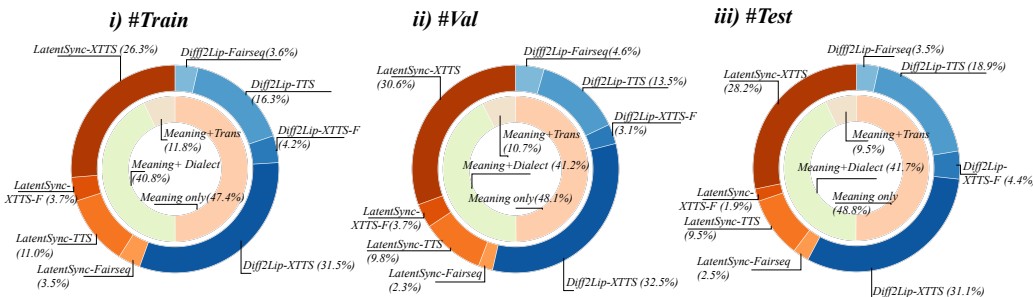

Figure 2: Dataset distribution for i) *Train*, ii) *Val* and iii) *Test* split. The outer layer shows the split between various combinations of Text-to-Speech and Lip-Sync models used for audio-visual manipulation. The middle layer shows the distribution of language in the original transcript, which is either *Ar* (Arabic) or *CSW* (Code-Switched English-Arabic). The inner layer shows the distribution of different operations applied to the original transcripts, "meaning only", "dialect+meaning", and "meaning + translation" (For fine-grained detail about what they entail, refer to Table 16 in Appendix.)

Table 1: **Details for publicly available deep-fake datasets in chronological ascending order.** Cla: Binary classification, SL: Spatial localization, TL: Temporal localization, FS: Face swapping, RE: Face reenactment, TTS: Text-to-speech, VC: Voice conversion.

| Dataset | Year | Tasks | Manip. Modality | Method | #Total | Multilingual | Code Switching |
|---|---|---|---|---|---|---|---|
| Google DFD (Nick & Andrew, 2019) | 2019 | Cla | V | FS | 3,431 | ✗ | ✗ |
| DFDC (Dolhansky et al., 2020b) | 2020 | Cla | AV | FS | 128,154 | ✗ | ✗ |
| DeeperForensics (Jiang et al., 2020) | 2020 | Cla | V | FS | 60,000 | ✗ | ✗ |
| Celeb–DF (Li et al., 2020b) | 2020 | Cla | V | FS | 6,229 | ✗ | ✗ |
| KoDF (Kwon et al., 2021) | 2021 | Cla | V | FS/RE | 237,942 | ✗ | ✗ |
| FakeAVCeleb (Khalid et al., 2022) | 2021 | Cla | AV | RE/FS | 25,500+ | ✗ | ✗ |
| ForgeryNet (He et al., 2021) | 2021 | SL/TL/Cla | V | Random FS/RE | 221,247 | ✗ | ✗ |
| ASVSpoof2021DF (Liu et al., 2023) | 2021 | Cla | A | TTS/VC | 593,253 | ✗ | ✗ |
| LAV–DF (Cai et al., 2022) | 2022 | TL/Cla | AV | Content-driven RE/TTS | 136,304 | ✗ | ✗ |
| DF–Platter (Narayan et al., 2023) | 2023 | Cla | V | FS | 265 756 | ✗ | ✗ |
| AV-1M (Cai et al., 2023a) | 2023 | TL/Cla | AV | Content-driven RE/TTS | 1,146,760 | ✗ | ✗ |
| PolyGlotFake (Hou et al., 2024) | 2024 | Cla | AV | RE/TTS/VC | 15,238 | ✓ | ✗ |
| Illusion (Thakral et al., 2025) | 2025 | Cla | AV | FS/RE/TTS | 1,376,371 | ✓ | ✗ |
| **ArEnAV (Ours)** | **2025** | **Cla/TL** | **AV** | **Content Driven RE/TTS/VC** | **387,072** | ✓ | ✓ |

## 3 ARENAV DATASET

ArEnAV is a large-scale audio-visual deepfake dataset specifically focused on Arabic–English CSW. Comprising approximately 765 hours of video data sourced from 8,809 unique YouTube videos, ArEnAV establishes itself as the first and most extensive benchmark for multilingual deepfake detection (see Table 1 for dataset comparison). The dataset is constructed to preserve the original identity and environmental context of the source videos while systematically manipulating the semantic content to introduce Arabic-English CSW. Following the taxonomy proposed by Cai et al. (2024a), ArEnAV includes three manipulation strategies: **Fake Audio & Fake Video**: Both audio and visual content are synthetically generated, simulating complete audiovisual deepfakes. **Fake Audio & Real Video**: The audio track is manipulated to introduce anti-semantic and CSW content while maintaining the original visual content. **Real Audio & Fake Video**: The original audio is retained, while facial movements and lip synchronization are altered to create visually deceptive content.

### 3.1 DATA COLLECTION

We use the YouTube video links from VisPer's Arabic Train subset (Narayan et al., 2024). We chose VisPer because it is the largest publicly available non-English audio-visual corpus, with over 1,200 hours of Arabic alone. Its 200-keyword crawler ("interview,", "tutorial," etc.) pulls videos spanning talk shows, vlogs, documentaries, and lectures, mirroring the broad-coverage strategy that is required for a fair and diverse representation of a culture-specific deepfake dataset. We first run a scene change detection model to split the video into clips, and then we use Yolo-v5 to obtain the faces in each frame as well as track them across frames. Since we did not have ground truth for transcripts for VisPER, we surveyed state-of-the-art Automatic Speech Recognition (ASR) models for Arabic, based on the Arabic ASR Leaderboard on HuggingFace. Following a qualitative comparison, we finalized the Whisper-v2 (finetuned on English-Arabic data) for our method, with the default output language set to Arabic. Following the transcripts, we apply Forced Alignment between the audio and text, using a multilingual wav2vec2 model (Baevski et al., 2020) supporting both Arabic and English. This provides us with word-level timestamps for code-switched Arabic and English data.

### 3.2 DATA GENERATION PIPELINE

The data generation pipeline roughly consists of three stages: transcript manipulation, audio generation, and video generation. First, we apply controlled modifications to the transcript. Secondly, we synthesize new audio for the altered transcript while preserving the speaker's voice characteristics. Finally, we render a lip-synced video that matches the new audio, producing a realistically manipulated video clip. We detail each stage as follows.

#### 3.2.1 TRANSCRIPT MANIPULATION

We leverage GPT-4.1-mini (OpenAI, 2025) to perform content-driven modifications of our multilingual transcripts. We define eight distinct transcript change modes that span both code-switched and Arabic-only contexts, allowing fine-grained control over how the transcript is altered. These modes include three main operations: first, *meaning only*, which only involves changing the meaning of the word and keeping the language as it is, second, *meaning + dialect*, which involves changing the meaning of the word and changing its language to another Arabic variant (either MSA or any dialect), and lastly, *meaning + translation*, which asks the model to change the meaning of the word, and then translate it to English. Table 2 summarizes the eight modification modes with their intended effect.

Table 2: Transcript manipulation rules in ArEnAV for Arabic (AR) and English (EN) words.

| # | Original Transcript | Original Word | Inserted Word | Operation |
|---|---|---|---|---|
| 1 | CSW | EN | EN | Change meaning only (keep English) |
| 2 | CSW | AR | AR | Change meaning only (keep Arabic variant) |
| 3 | CSW | AR | AR | Change meaning + change Arabic variant |
| 4 | CSW | AR | AR/EN | multi-op; When 2-3 ops → edit 1 EN and 1-2 AR words |
| 5 | Arabic | AR | AR | Change meaning only (keep Arabic variant) |
| 6 | Arabic | AR | AR | Change meaning + change Arabic variant |
| 7 | Arabic | AR | EN | Change meaning + change language to English |
| 8 | Arabic | AR | AR/EN | multi-op; Apply all operations on Arabic words |

By categorizing edits in this way, we ensure a controlled and diverse set of manipulations ranging from subtle word substitutions to introducing or removing CSW instances. Due to the effectiveness of few-shot prompting, we prompt GPT-4.1-mini with 15 examples, explaining various kinds of transitions and possible changes. Examples of original and augmented transcripts achieved by these manipulation rules are shown in Appendix A.5. We provide the prompt in Appendix A.6. We report text manipulations distributions as follows: replacement (94.6%), insertion (5.1%) and deletion (0.3%). These distributions reflect GPT's manipulation choices, which were not manually enforced.

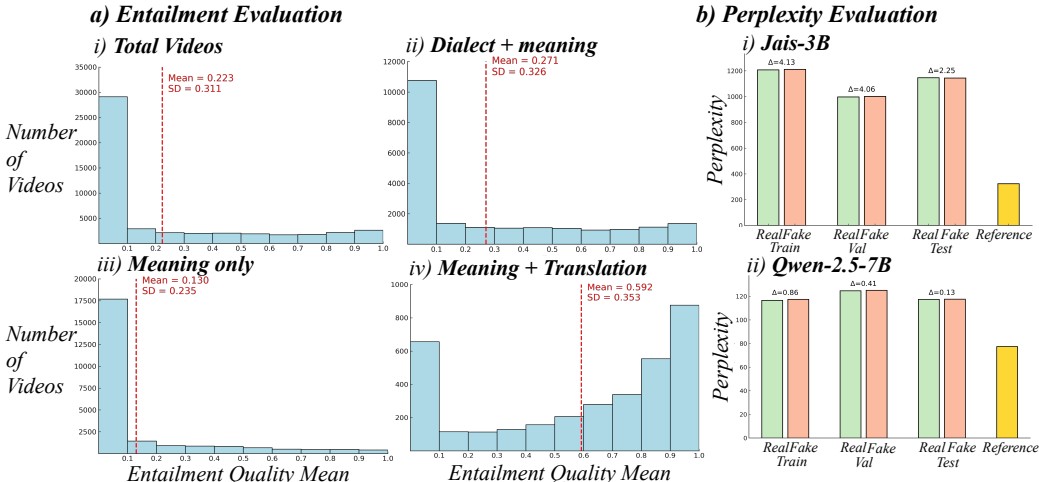

Figure 3: a) Entailment distribution over i) All change modes, ii) Dialect + meaning, iii) Meaning only, and iv) Meaning + translation. b) Perplexity Evaluation distribution among dataset splits, showing perplexity calculated using i) Jais-3B, an Arabic-English LLM, and ii) Qwen-2.5-7B. b) Perplexity calculated using i) Jais-3B ii) Qwen-2.5-7B. Reference in both shows the perplexity calculated on an Arabic-English CSW text dataset (SDAIANCAI, 2025).

***Transcript Quality:***

To quantify the impact of our LLM-based manipulations, we employ two complementary metrics: ***Bidirectional Entailment Quality Mean***: the average of Real→Fake and Fake→Real NLI entailment scores (1.0 = full semantic entailment; 0.0 = direct contradiction) and ***Perplexity***: how well a language model predicts a transcript (lower = more fluent/natural). Table 3a shows the distribution of entailment quality means over different types of perturbations. In every subset, a large fraction of samples lies below the 0.5 threshold, and many even in the contradiction zone, demonstrating that our pipeline reliably injects semantic change regardless of language or dialect.

Figure 3b reports average perplexities on real versus fake transcripts under two open-source LLMs; Jais-3B (Sengupta et al., 2023) and Qwen-2.5-7B (Qwen et al., 2025), across the data splits. The minimal difference in perplexity shows that our fake transcripts remain fluent and natural, despite major changes in meaning. This balance between altered content and surface-level fluency is essential for generating effective audio-visual deepfakes

### 3.2.2 AUDIO GENERATION

The next step involves generating a synthetic audio track that precisely follows the edited transcript while maintaining the voice characteristics of the original on-screen speaker. Initially, we segment the audio into clean speech and background noise using a Denoiser (Defossez et al., 2020). Conventional zero-shot voice cloning systems, such as YourTTS (Casanova et al., 2023), exhibit strong performance in English but struggle with Arabic phonetics and cross-lingual synthesis. To address this, we employ four targeted cloning strategies: a) **XTTS-v2 (Casanova et al., 2024)**: A multilingual, zero-shot TTS model natively supporting Arabic, English, and code-switching. b) **XTTS-v2 (Casanova et al., 2024) + OpenVoice-v2 (Qin et al., 2024)**: When a reference voice sample is available, we achieve higher fidelity by generating the utterance with XTTS-v2 and performing speaker conversion via OpenVoice-v2. c) **Fairseq Arabic TTS (Ott et al., 2019) + OpenVoice-v2 (Qin et al., 2024)**:

For fully Arabic sentences, we generate audio with the Fairseq Arabic TTS, followed by speaker conversion using OpenVoice-v2. d) **GPT-TTS (OpenAI, 2023) + OpenVoice-v2 (Qin et al., 2024)**: We randomly select one voice out of 29, generate the sentence, and then convert the audio to the target speaker's voice with OpenVoice-v2.

The audio-generation flow depends on the edit type. For *insert* or *replace* operations, we regenerate the complete sentence and validate the generated audio using Whisper-Turbo (Radford et al., 2022), retaining only samples that exactly match the intended transcript. This step ensures intelligibility and accurate timestamp alignment for splicing the segment into the original audio. If validation fails, we discard the sample. For a *delete* operation, we remove speech segments entirely, preserving only background noise. Finally, after each edit, we normalise the loudness of the manipulated segment relative to the original audio and recombine it with the extracted environment noise.

*Audio Quality:*

Table 3 presents the comparison of audio quality for ArEnAV based on speaker similarity, signal quality and distribution realism with existing audio-visual deepfake datasets. We report speaker encoder cosine similarity (SECS), Signal-to-Noise (SNR) and Fréchet audio distance (FAD) for recent Audio-Visual datasets. SECS measures the speaker's voice similarity between a generated clip and the real reference (range $[-1, 1]$, higher is better), while FAD evaluates

Table 3: Audio quality comparison across different datasets.

| Dataset | Language | SECS↑ | SNR(dB)↑ | FAD↓ |
|---|---|---|---|---|
| FakeAVCeleb | English | 0.543 | 2.16 | 6.598 |
| LAV-DF | English | 0.984 | 7.83 | 0.306 |
| AV-Deepfake1M | English | 0.991 | 9.39 | 0.088 |
| **ArEnAV** | **Arabic, English** | **0.990** | **7.65** | **0.140** |

the distributional distance between the generated audio and real audio (lower is better). The metrics combined indicate that ArEnAV has high-quality audio samples.

## 3.3 VISUAL MANIPULATION

For video generation, after extensive experimentation, we chose two diffusion-based lip-sync approaches: Diff2Lip (Mukhopadhyay et al., 2023) and LatentSync (Li et al., 2025). Both of these models perform high-quality zero-shot lip-sync and are open-sourced. Using the newly generated audio and the original video's frames we generate the fake frames. For *replace* and *insert* word operations, we generate the fake frames for the new word, and for *delete* word operations, we generate a face with closed lips i.e., without audio.

*Visual Quality:* To evaluate visual quality, we use three standard metrics: Peak Signal-to-Noise Ratio (PSNR), Structural Similarity Index (SSIM), and Fréchet Inception Distance (FID). Table 4 presents PSNR, SSIM, and FID results for the ArEnAV dataset. PSNR and SSIM measure pixel-level and structural similarity, respectively, between fake and original frames (higher is better) (ArEnAV lies nearby AV-1M). FID assesses realism by comparing the distributions of fake and real frames in a learned image feature space (lower is better) (ArEnAV slightly more than AV-1M). These scores highlight that ArEnAV attains higher / comparable visual quality compared to other deepfake datasets.

Table 4: Visual quality comparison across different datasets.

| Dataset | PSNR(dB)↑ | SSIM↑ | FID↓ |
|---|---|---|---|
| FF++ | 24.40 | 0.812 | 1.06 |
| DFDC | — | — | 5.69 |
| FakeAVCeleb | 29.82 | 0.919 | 2.29 |
| LAV-DF | 33.06 | 0.898 | 1.92 |
| AV-Deepfake1M | 39.49 | 0.977 | 0.49 |
| **ArEnAV** | **37.70** | **0.971** | **0.68** |

**Real Perturbations:** To mimic real-life video scenarios better, we add localized perturbations to both the real and the fake videos. We apply 15 different visual filters (eg, salt-pepper noise and camera shaking) and 10 different audio manipulations (eg, time-stretching, random loudness and pitch). For each video, we randomly sample one to three instances for visual perturbations and one to two instances for audio perturbations. Perturbation details are mentioned in Appendix A.4.

## 3.4 USER STUDY

To investigate whether humans can identify deepfakes in ArEnAV, we conducted a user study with 19 participants, out of which, 15 are native Arabic speakers, and 4 have basic knowledge of Arabic. We randomly sampled 20 videos, with either 0 or 1 manipulation. ***Instructions for User Study:***

Table 5: Detection and Localization results from our User Study.

| Method | Acc. | AP@0.1 | AP@0.5 | AR@1 |
|---|---|---|---|---|
| ArEnAV | 60.00 | 8.35 | 0.79 | 1.38 |

Each participant was asked to 1) watch the video, and 2)
answer 3 questions, i) Is the video real of fake, ii) If it is fake, localize where they think the fake
region is, iii) Whether the given video contains Arabic-English code-switching or not, iv) Give a
reason for labelling the video (if they have) as a deepfake.

The results in Table 5 reaffirm our hypothesis that identifying
audiovisual deepfakes in multilingual (specially CSW) and
multimodal settings is a non-trivial task, as even humans
achieve only 60% accuracy, while it is even harder to localize
the deepfakes, with AP@0.5 at 0.79. Further, Table 6 shows
the primary reasons why people classified the videos as fake.
We report that 85% of the users fail to identify deepfakes
when the manipulation happens in the English word, in the
CSW video, which can be attributed to a higher quality of
voice cloning in English as well as the natural change in
tone when a person code-switches, which makes it harder
to detect. Further, localization is very tough due to the very

Table 6: Distribution of top reasons for
predicting a video as Fake in our User
Study.

| Reason | Percentage (%) |
|---|---|
| Unintelligible speech (weird audio) | 36.5 |
| Video/audio mismatch (lip sync is off) | 25.1 |
| Audio sounds artificial | 24.7 |
| Video looks artificial | 8.7 |
| Code-switching is unnatural | 3.0 |
| Incoherent sentence | 1.9 |

high quality of lip-sync with diffusion models, as shown in Table 6, where the video being the reason
for fake classification is only 8.7%.

## 3.5 DATASET STATISTICS

Table 7: Data distribution in ArEnAV and comparison with other multilingual datasets.

| Subset | #Unique Videos | #Real Videos | #Fake Videos | #Non-English Clips | #CSW Videos | #Arabic Videos | Arabic Variants |
|---|---|---|---|---|---|---|---|
| PolyGlotFake (Hou et al., 2024) | 766 | 766 | 14,472 | **11,941** | 0 | 1,403 | NA |
| Illusion (Thakral et al., 2025) | – | 141,440 | 1,234,931 | **4,385** | 0 | – | NA |
| **ArEnAV-Train** | 6,117 | 67,600 | 202,800 | 270,400 | 69,544 | 200,856 | Egyptian, |
| **ArEnAV-Validation** | 876 | 9,560 | 28,680 | 38,240 | 10,416 | 27,824 | MSA, |
| **ArEnAV-Test** | 1,816 | 19,608 | 58,824 | 78,432 | 19,832 | 58,600 | Levantine, Gulf |
| **ArEnAV (total)** | **8,809** | **96,768** | **290,304** | **387,072** | **99,792** | **287,280** | - |

Table 7 compares ArEnAV with other multilingual deepfake detection datasets. Existing multilingual
datasets like PolyGlotFake (Hou et al., 2024) and Illusion (Thakral et al., 2025) have significantly
smaller multilingual content, containing limited Arabic data (1,400 Arabic videos in PolyGlotFake
and minimal in Illusion across 26 languages). ArEnAV includes 387k videos sourced from 8,809
unique YouTube videos, totaling over 765 hours. Videos average approximately 7.7 sec each, with
train, val, and test splits created via multilabel stratified sampling in a 7:1:2 ratio, ensuring no overlap.

Table 8: Fake segment duration comparison between ArEnAV and AV-1M.

| Dataset | Mean (s) | Median (s) | Minimum (s) | Maximum (s) | Video-length (s) | Fake ratio (%) | Relative length (×) |
|---|---|---|---|---|---|---|---|
| **ArEnAV** | **0.696** | **0.625** | **0.02** | **6.16** | **5.97** | **12.1** | **2.1** |
| AV-1M | 0.326 | — | — | — | 9.07 | 3.7 | 1.0 |

**Fake Region Comparison::** In Table 8, we summarize forged-segment duration statistics for
ArEnAV and AV-1M. The Table highlights substantially longer and proportionally larger forged spans
in ArEnAV (Fake segments are 2.1 times longer relative to AV-1M), confirming that performance
drops stem from the intrinsic difficulty of detecting linguistically precise intra-utterance manipulations
rather than from short spans.

**Computational Cost:** We spent around 50 GPU hours to generate the real transcript using Whisper-
Large-V2 (Radford et al., 2022), 200 dollars worth of OpenAI credits, to generate fake transcripts and
Text-to-Speech model, TTS-1 (OpenAI, 2023), and 650 GPU hours for video generation. Overall, we
needed 800 GPU hours to generate AvEnAV with NVIDIA RTX-*6000* GPUs.

## 4 BENCHMARK AND METRICS

We organize the data into *train*, *validation*, and *test* split. We use multilabel stratified sampling to
divide the data in equal proportions based on the method type, the change mode, and the ground truth
language. We also show evaluation on two subsets, *subset V*, which excludes videos with audio-only

Table 9: Temporal localization results on the test set of ArEnAV.

| Set | Method | Mod. | AP@0.5 | AP@0.75 | AP@0.9 | AP@0.95 | AR@50 | AR@30 | AR@20 | AR@10 | AR@5 |
|---|---|---|---|---|---|---|---|---|---|---|---|
| Full dataset | Meso4 | V | 0.02 | 0.01 | 0.00 | 0.00 | 0.09 | 0.09 | 0.09 | 0.09 | 0.09 |
| – | MesoInception | V | 0.56 | 0.18 | 0.04 | 0.01 | 4.11 | 4.11 | 4.11 | 4.11 | 4.08 |
| – | Xception | V | 22.50 | 10.26 | 2.29 | 0.58 | 19.13 | 19.13 | 19.13 | 19.13 | 19.13 |
| – | BA-TFD (ZS) | AV | 0.17 | 0.01 | 0.00 | 0.00 | 9.72 | 5.20 | 3.07 | 1.46 | 0.73 |
| – | BA-TFD+ (ZS) | AV | 0.11 | 0.00 | 0.00 | 0.00 | 5.77 | 2.95 | 2.09 | 0.87 | 0.37 |
| – | BA-TFD | AV | 2.42 | 0.55 | 0.01 | 0.00 | 22.30 | 10.31 | 3.41 | 2.54 | 1.67 |
| – | BA-TFD+ | AV | 3.74 | 1.10 | 0.06 | 0.01 | 30.75 | 9.42 | 4.55 | 3.05 | 1.83 |
| Set V | Meso4 | V | 0.02 | 0.01 | 0.00 | 0.00 | 0.10 | 0.10 | 0.10 | 0.10 | 0.10 |
| – | MesoInception | V | 0.83 | 0.27 | 0.05 | 0.01 | 5.56 | 5.56 | 5.56 | 5.56 | 5.53 |
| – | Xception | V | 32.76 | 14.48 | 3.30 | 0.81 | 27.78 | 27.78 | 27.78 | 27.78 | 27.78 |
| – | BA-TFD (ZS) | AV | 0.12 | 0.00 | 0.00 | 0.00 | 8.44 | 4.34 | 2.44 | 1.13 | 0.49 |
| – | BA-TFD+ (ZS) | AV | 0.07 | 0.00 | 0.00 | 0.00 | 4.69 | 2.39 | 1.65 | 0.69 | 0.29 |
| – | BA-TFD | AV | 3.65 | 0.25 | 0.01 | 0.00 | 25.31 | 9.03 | 3.64 | 2.34 | 1.64 |
| – | BA-TFD+ | AV | 5.65 | 1.89 | 0.08 | 0.02 | 31.09 | 13.21 | 5.91 | 3.05 | 2.05 |
| Set A | Meso4 | V | 0.02 | 0.01 | 0.00 | 0.00 | 0.08 | 0.08 | 0.08 | 0.08 | 0.08 |
| – | MesoInception | V | 0.38 | 0.09 | 0.01 | 0.00 | 3.25 | 3.25 | 3.25 | 3.25 | 3.22 |
| – | Xception | V | 14.72 | 3.92 | 0.29 | 0.09 | 11.78 | 11.78 | 11.78 | 11.78 | 11.78 |
| – | BA-TFD (ZS) | AV | 0.23 | 0.01 | 0.00 | 0.00 | 12.14 | 6.46 | 3.85 | 1.83 | 0.95 |
| – | BA-TFD+ (ZS) | AV | 0.14 | 0.01 | 0.00 | 0.00 | 7.32 | 3.79 | 2.69 | 1.13 | 0.48 |
| – | BA-TFD | AV | 3.21 | 0.60 | 0.02 | 0.00 | 24.45 | 9.26 | 4.15 | 2.61 | 1.93 |
| – | BA-TFD+ | AV | 4.35 | 1.10 | 0.10 | 0.00 | 28.35 | 11.23 | 4.85 | 3.11 | 2.00 |

manipulation, and *subset A*, which excludes videos with visual-only manipulations. We evaluate models on two tasks, **temporal localization** and **detection** of audio-visual deepfakes. We use average precision (AP) and average recall (AR) metrics as prior works (He et al., 2021; Cai et al., 2022; 2023a) for temporal localization. For the task of deepfake detection, we use the standard evaluation protocol (Rossler et al., 2019; Dolhansky et al., 2020b; Cai et al., 2023a) to report video-level accuracy (Acc.) and area under the curve (AUC).

**Implementation Details:** We benchmark **temporal detection** using SOTA models: Meso4, MesoInception4, Xception, BA-TFD, and BA-TFD+. BA-TFD and BA-TFD+ (Cai et al., 2023b) are evaluated in their original configurations, both in a zero-shot setting (pre-trained on AV-1M; (Cai et al., 2023a)) and after fine-tuning on our dataset. For image-based classifiers: Meso4, MesoInception4 (Afchar et al., 2018); and Xception (Chollet, 2017), we aggregate frame-level predictions to segments following Cai et al. (2023a). For benchmarking **deepfake detection**, image-based models (Meso4, MesoInception4, and Xception) are trained on video frames with corresponding labels, and predictions are aggregated to video-level using max voting, as suggested by Cai et al. (2023a). Additionally, we assess zero-shot performance of LLM-based models, VideoLLaMA2 and VideoLLaMA2.1-AV (Zhang et al., 2023), prompting them to produce a confidence score indicating the likelihood of a video being a deepfake. We include an audio-only baseline, XLSR-Mamba (Xiao & Das, 2025), the best open-source audio deepfake detection model on Speech DF Arena (Face, 2025), evaluating it both in zero-shot mode (pre-trained on ASVSpoof-2019 (Wang et al., 2020)) and after training with video-level labels from our dataset. BA-TFD and BA-TFD+ (Cai et al., 2022) are also evaluated using segmentation proposals treated as frame-level predictions and aggregated by max-voting, both pre-trained on AV-1M and fine-tuned on our dataset. For all finetuning runs, we subsample the frames so as to remove class imbalance.

## 5 RESULTS AND ANALYSIS

**Audio-Visual Temporal Deepfake Localization.** The results for temporal localization are shown in Table 9. SOTA methods show significantly lower performance on ArEnAV as compared to other localization datasets (refer to Table 11a). BA-TFD and BA-TFD+, pretrained on AV-1M, show a drop in performance of more than 35% for AP@0.5 threshold, compared to evaluation on AV-1M. The image-based models, Meso4 and MesoInception4, also provide low performance, which can be attributed to the use of diffusion-based lip-sync models, which have been overlooked in previous data generation pipelines (Cai et al., 2023a;b). Through this benchmark, we claim that the highly realistic multimodal multilingual code-switched fake content in ArEnAV will open an avenue for further research on temporal multilingual deepfake localization methods.

Table 10: Deepfake detection results on the test set of ArEnAV.

| Label Access For Training | Pretraining Data | Methods | Mod. | Fullset AUC | Acc. | Subset V AUC | Acc. | Subset A AUC | Acc. |
|---|---|---|---|---|---|---|---|---|---|
| Zero-Shot | ASVSpoof-19 | XLSR-Mamba | A | 39.19 | 52.77 | 52.73 | 40.68 | 52.50 | 42.59 |
| - | Internet Scale | Video-LLaMA (7B) | V | 51.48 | 26.29 | 51.47 | 34.21 | 51.43 | 34.18 |
| - | Internet Scale | Video-LLaMA (7B) | AV | 48.79 | 59.29 | 48.71 | 55.37 | 48.86 | 55.26 |
| - | AV-1M | BA-TFD | AV | 61.73 | 26.00 | 66.42 | 34.07 | 59.36 | 33.97 |
| - | AV-1M | BA-TFD+ | AV | 60.96 | 25.84 | 64.49 | 34.28 | 59.44 | 33.80 |
| Video Level | ArEnAV | XLSR-Mamba | A | 73.00 | 61.00 | 57.47 | 66.16 | 86.33 | 78.00 |
| - | ArEnAV | Meso4 | V | 49.30 | 75.00 | 49.15 | 66.67 | 49.30 | 66.67 |
| - | ArEnAV | MesoInception4 | V | 50.34 | 46.23 | 50.28 | 47.48 | 50.35 | 47.67 |
| - | ArEnAV | Xception | V | 50.05 | 75.00 | 49.90 | 66.67 | 50.32 | 66.67 |
| Frame level | ArEnAV | Meso4 | V | 49.55 | 26.60 | 49.60 | 34.40 | 49.53 | 34.36 |
| - | ArEnAV | MesoInception4 | V | 51.14 | 41.25 | 50.77 | 51.84 | 45.28 | 44.09 |
| - | ArEnAV | Xception | V | 74.21 | 52.09 | 85.36 | 67.22 | 68.59 | 51.70 |
| - | AV-1M & ArEnAV | BA-TFD | AV | 75.91 | 44.31 | 77.64 | 58.29 | 72.21 | 45.21 |
| - | AV-1M & ArEnAV | BA-TFD+ | AV | 79.97 | 27.44 | 84.20 | 36.47 | 72.89 | 34.56 |

Table 11: (a): Temporal localization comparison on ArEnAV, AV-1M and LAVDF. (b): Cross-Dataset comparison (% AUC) of recent SOTA models.

(a) Cross-dataset Deepfake Localization.

| Method | Dataset | AP@0.5 | AP@0.95 | AR@50 | AR@10 |
|---|---|---|---|---|---|
| BA-TFD | LAV-DF | 79.15 | 0.24 | 64.18 | 58.51 |
| | AV-1M | 37.37 | 0.02 | 45.55 | 30.66 |
| | **ArEnAV** | **2.42** | **0.01** | **22.30** | **2.54** |
| BA-TFD+ | LAV-DF | 96.30 | 4.44 | 80.48 | 78.75 |
| | AV-1M | 44.42 | 0.03 | 48.86 | 34.67 |
| | **ArEnAV** | **3.74** | **0.04** | **30.75** | **3.05** |

(b) Cross-dataset deepfake detection. P: DFDC-P set.

| Method | Venue | ArEnAV | DFDC | FF++ | CelebDF |
|---|---|---|---|---|---|
| Capsule-v2 | ICASSP-19 | 49.15 | – | 93.11 | – |
| Face-X-Ray | CVPR-20 | 55.56 | 80.92 | 98.52 | 80.58 |
| LipForensics | CVPR-21 | 49.76 | 73.50 | 97.10 | 82.40 |
| M2TR | ICMR-22 | 50.12 | – | 99.92 | – |
| LAA-Net | CVPR-24 | 50.04 | 86.94(P) | 99.96 | – |
| ForensicsAdaptor | CVPR-25 | 50.58 | 88.70 | – | 94.00 |

**Audio-Visual Deepfake Detection.** The detection results are in Table 10. Image based models, that have access to video-level labels only, perform considerably worse, except XLSR-Mamba, which is designed to be trained on video-level labels for audio-deepfake detection. The best performing model is BA-TFD, pretrained on AV-1M and then further fine-tuned on our dataset, with AUC Score of 82% on the full subset. We also evaluate models on subsets V and A, as described in the implementation details. The audio-only model, XLSR-Mamba, performs better in the Audio-only *subset A*, while the image-only models perform better on *Subset V* for frame-level labels, compared to the *fullset*. XLSR-Mamba performs relatively worst when the audio is code-switched, compared to only Arabic.

**Cross-Dataset Comparison for Deepfake Localization.** Table 11a shows the performance of BA-TFD and BA-TFD+ (Pretrained on AV-1M) on LAVDF, AV-1M and ArEnAV datasets. Both models perform significantly worse on ArEnAV, highlighting the poor generalizability (while it generalizes to LAV-DF) in multilingual and code-switching settings. BA-TFD and BA-TFD+ fail to generalize effectively, as the pretrained audio and video encoders struggle with out-of-distribution data encountered in both modalities of ArEnAV.

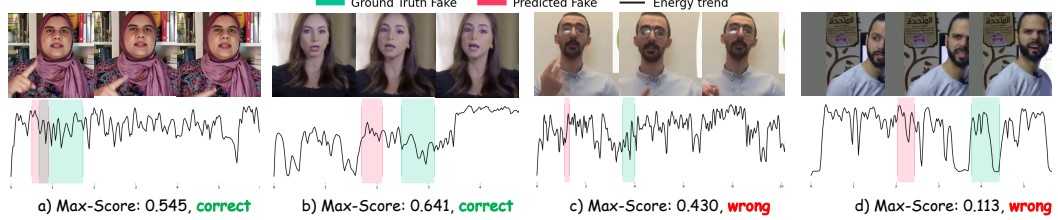

**Ground Truth Fake** **Predicted Fake** **Energy trend**

a) Max-Score: 0.545, *correct*  b) Max-Score: 0.641, *correct*  c) Max-Score: 0.430, *wrong*  d) Max-Score: 0.113, *wrong*

Figure 4: Different output cases from evaluation of BA-TFD+ after finetuning on ArEnAV. Here, the ground truth of all samples is **FAKE**. Max-Score refers to the maximum score assigned to a candidate range during prediction. Correct means that the predicted class matches the ground truth class. The green region refers to the ground truth fake-segment, and the red region refers to the predicted fake-segment, based on the Max Score. a) Shows the model predicting the correct class, along with some overlap with the ground truth fake segment. b) Shows the model predicting the correct class, but with no overlap. c) and d) Show the model predicting the wrong class in the output.

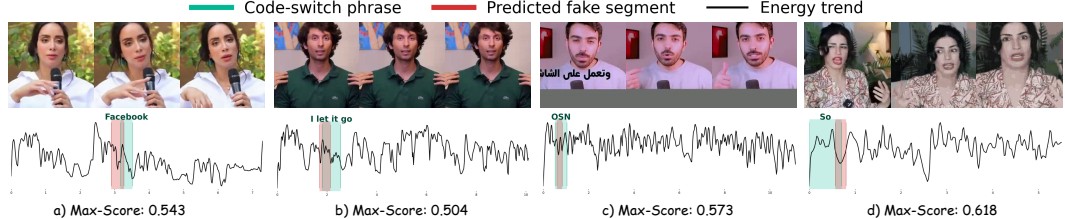

Figure 5: Different output cases from evaluation of BA-TFD+ after finetuning on ArEnAV. Here, the ground truth of all samples is **REAL**. Max-Score refers to the maximum score assigned to a candidate range during prediction. The green region refers to the code-switching region in the real video, and the red region refers to the predicted fake-segment. a),b),c),d) All samples show that the model misclassifies real videos as fakes, specifically at the code-switching regions.

**Cross-Dataset Comparison for Deepfake Detection.** Table 11b shows the cross dataset performance of recent SOTA deepfake detection models, including Capsule-v2 (Nguyen et al., 2019), Face-X-Ray (Li et al., 2020a), LipForensics (Haliassos et al., 2021) and M2TR (Wang et al., 2022). All models were pretrained on FaceForensics++ (Rössler et al., 2019). While models show great cross-dataset performance on DFDC and CelebDF, they fail to perform better than guessing (50% AUC) on ArEnAV. Even recent SOTA models, such as ForensicsAdaptor (CVPR-25) (Cui et al., 2025b) and LAA-Net (CVPR-24) (Nguyen et al., 2024), fail to generalize. The demographic and linguistic homogeneity of existing datasets (FF++, CelebDF, DFDC) limits model robustness. By incorporating multilingual audio and broader participant diversity, our dataset demonstrates why architectures must be designed to generalize beyond those biases.

## 6 QUALITATIVE ANALYSIS

Figures 4 and 5 show qualitative examples for different output cases of BA-TFD+, finetuned on ArEnAV. Figure 4 illustrates model predictions on fake samples, showing four representative cases comparing predicted fake segments with ground-truth regions. In some cases, the model correctly identifies the manipulated class with good temporal overlap, while in others, the predictions are either misaligned or incorrect. Importantly, these outputs demonstrate that the model's behavior is not influenced by any boundary-level artifacts or splicing cues introduced during data generation. Further, spectrograms from BA-TFD+ outputs show no abrupt energy shifts or discontinuities near edit boundaries, indicating the absence of splice artifacts at the boundaries of manipulated content.

Figure 5 shows the model's predictions on real code-switched videos. Here, the model frequently misclassifies real videos as fake, primarily due to the presence of code-switching between Arabic and English. The high predicted fake scores at these regions indicate that the model confuses natural linguistic transitions with synthetic inconsistencies. Together, these qualitative results confirm that the challenge in ArEnAV arises from the intrinsic complexity of code-switching rather than from generation artifacts.

## 7 CONCLUSION

This paper presents ArEnAV, a large multilingual and the first code-switching audio-visual dataset for temporal deepfake localization and detection. The comprehensive benchmark of the dataset utilizing SOTA deepfake detection and localization methods, indicates a significant drop in performance compared to previous monolingual datasets, indicating that the proposed dataset is an important asset for building the next generation of multilingual deepfake localization methods.

*Limitations*. Similar to other deepfake datasets, ArEnAV exhibits a misbalance in terms of the number of fake and real videos. Due to the limited performance of current SOTA Active-Voice-Detection (Whisper v2) models on Arabic (compared to English), the data generation pipeline can result in a few noisy transcripts. Due to limited instruction following in code-switching scenarios, LLMs might not produce the desired results, as visible in Figure 3 "Meaning + Translation Scenario". Compared to other subsets, Chat-GPT often fails to follow both instructions, making real and fake transcripts too similar and not always changing their meaning. Also, the dataset is currently limited to two languages only, where we hope to motivate further research in this direction.

**Broader Impact.** ArEnAV's diverse and realistic English-Arabic fake videos will support the development of more robust audio-visual deepfake detection and localization models, better equipped to handle code-switched speech and real-world multilingual scenarios.

## 8 ETHICS STATEMENT

Our work on ArEnAV raises important ethical considerations, especially given the sensitivity of deep-fake research. The dataset is built from publicly available YouTube content, in line with established practices in benchmarks. Use of such material for non-commercial research is covered under fair use, and access to ArEnAV is gated by a strict End-User License Agreement (EULA)(Section A.7). Below, we detail all the ethical considerations regarding our work:

***Use of YouTube Videos*** : The ethical foundation of our data collection does not rely on VisPER but on established practices in prior peer-reviewed datasets such as LRS3-TED (Afouras et al., 2018b), VoxCeleb2 (Chung et al., 2018b), and AVSpeech (Ephrat et al., 2018a), which employ the same keyword-search and face-detection pipeline. We build our dataset from public YouTube videos under the research-focused "fair use" exception established in peer-reviewed work (e.g., Zhu et al. (2024)). Access is granted only after users agree to our EULA, which lists the following rules and regulations:

- Access will be granted only to researchers who supply their university IRB application ID, and every project member must use an individual account, safeguarding traceability and preventing misuse.
- Users are eligible to conduct independent research at their respective institutions and the Institution accepts responsibility for its Authorized Investigators' actions related to the use of ArEnAV.
- Limits use to academic, non-commercial, not-for-profit research and education.
- Authorizes licensors to modify the data or license at any time and prohibits licensees from altering the database.
- Forbids any use that could cause subjects embarrassment or mental anguish.

This approach accords with established practice across the community, as evidenced by DF40 (Yan et al., 2024), which draws real videos and images from FaceForensics++ (Rössler et al., 2019), Celeb-DF (Li et al., 2020b), CelebA (Liu et al., 2015), FFHQ (Karras et al., 2019), and VFHQ (Xie et al., 2022); DeepfakeBench (Yan et al., 2023), which relies on FaceForensics++ and Celeb-DF; FaceForensics++; Celeb-DF; FakeAVCeleb (Khalid et al., 2021), which builds on VoxCeleb2 (Chung et al., 2018a); AVLIPS (Liu et al., 2024), which sources from LRS3 (Afouras et al., 2018a) and FaceForensics++; and AV-1M (Cai et al., 2024b), which is derived from VoxCeleb2. Together, these measures and precedents demonstrate that curating public YouTube content for non-commercial scientific inquiry is a responsible and widely adopted practice.

***Face Detection techniques applied on videos***: We acknowledge the risks of working with videos that contain faces, but face detection is used only as a preprocessing step and not for identification. In line with recent peer-reviewed works, using videos containing faces, for different research problems that involve face detection as a common prior step, is a standard practice, e.g.: a) LRS3-TED (Afouras et al., 2018b), VoxCeleb1 (Nagrani et al., 2017), VoxCeleb2 (Chung et al., 2018b) have been used for Speaker Identification, Verification, Recognition, and further, for Deepfake benchmark creation (AV-1M (Cai et al., 2024a) and FakeAVCeleb (Khalid et al., 2022)) b) MultiTalk (Sung-Bin et al., 2024) uses videos from YouTube for Talking Head generation c) AVSpeech (Ephrat et al., 2018b) used for Speech Separation d) Hallo3 (Cui et al., 2025a) used for Portrait Image Animation.

Since it is impractical to get individual consent for open-source content, we mitigate misuse by requiring institutional IRB approval, individual researcher accounts for access and a removal mechanism to request the removal of personal content.

***Human Study***: Our human study followed university IRB guidelines: participants were over 18 years of age and were approached over email through connections in research groups within the affiliated universities of the authors. The participation was strictly voluntary and anonymous. All the details about the research project and conditions for participation in the study were clarified through an Explanatory Statement at the beginning of the user study form. Thus, the users consented to participate in the study by filling out and submitting the study form (Google form), and all material was screened to avoid disturbing content. No personal data were recorded and no compensation was provided.

## 9  REPRODUCIBILITY STATEMENT

Our data will be open-sourced. Data-generation code and evaluation scripts will be made public for various open-sourced models evaluated.

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

## A APPENDIX

### A.1 IDENTITY ANALYSIS IN ARENAV

We conducted a face analysis using InsightFace and DBSCAN to better understand potential identity overlaps across dataset splits. While each split (train, validation, and test) already contains independent YouTube video IDs, we found a 41.7% identity match between the combined train+val and test sets at a similarity threshold of 0.7.

To verify whether this overlap influences model performance, we evaluated BA-TFD and BA-TFD+ separately on overlapping and non-overlapping subsets. As shown in Table 12, the results remain consistent across both groups, indicating that identity overlap does not affect the models' behavior. We will release a separate identity-disjoint test set in the final version to further strengthen reproducibility and fairness.

Table 12: Performance of BA-TFD and BA-TFD+ models on ArEnAV test set. Each column shows Accuracy@0.5 and AUC, with the number of evaluated videos in parentheses.

| Model | Overall (78,400) | | Non-Overlapping (45,512) | | Overlapping (32,888) | |
|---|---|---|---|---|---|---|
| | Acc | AUC | Acc | AUC | Acc | AUC |
| **BA-TFD+** | 27.44 | 79.97 | 27.58 | 79.96 | 27.31 | 79.99 |
| **BA-TFD** | 44.31 | 75.91 | 44.35 | 75.96 | 44.27 | 75.85 |

### A.2 AFFECT OF LIMITED INSTRUCTION FOLLOWING OF GPT-4O-MINI FOR "MEANING + TRANSLATION" TASK:

In some cases, the "meaning + translation" mode produced sentences that were semantically similar to the originals. This can be improved by reprocessing the LLM outputs and filtering based on entailment or cosine distance to enforce greater semantic variation. However, this does not affect our main conclusions. The detection task in ArEnAV focuses on identifying audio-visual inconsistencies, not the extent of semantic change.

Only a small fraction of the data (less than 8,000 out of 280,000 samples, and about 2,000 out of 78k in the test set) (Figure 3a)) has an entailment quality mean > 0.8.

To reaffirm this, we compared the best-performing models across these subsets. As shown in Table 13, the performance remains consistent, indicating that these few cases do not lower the dataset's overall difficulty.

Table 13: Performance of BA-TFD and BA-TFD+ models on ArEnAV test set. Each column shows AUC scores.

| Model | Overall | entailment<0.8 | entailment>0.8 |
|---|---|---|---|
| **BA-TFD+** | 79.97 | 79.96 | 79.98 |
| **BA-TFD** | 75.91 | 75.92 | 75.89 |

### A.3 AFFECT OF DIFFERENT LIP-SYNC AND TTS MODELS ON THE PERFORMANCE:

Table 14 reports model accuracy across four TTS and 2 lip-sync generation methods (finetuned on ArEnAV). For audio-only detection, XLSR-Mamba achieves the highest accuracy with Fairseq-OpenVoice (95.8%), indicating that lower-quality TTS outputs are more easily detected. Higher-fidelity systems such as XTTS-OpenVoice and XTTS-v2 reduce detectability, suggesting improved synthesis quality. For audio-visual models (BA-TFD and BA-TFD+), both Diff2Lip and LatentSync yield similar accuracy levels, confirming stable and high-quality visual performance. Overall, the results show that better generation quality leads to lower detectability, aligning with our goal of creating challenging, realistic benchmarks.

Table 14: Accuracy across different Audio and Video generation models used in ArEnAV.

| Model Type | Audio Models (Subset A) | | | | Video Models (Subset-V) | |
|---|---|---|---|---|---|---|
| | Fairseq-OpenVoice | GPT-TTS | XTTS-OpenVoice | XTTS-v2 | Diff2lip | Latentsync |
| XLSR-Mamba | 95.8 | 78.8 | 19.4 | 77.9 | 38.6 | 42.8 |
| BA-TFD+ | 69.5 | 58.9 | 7.7 | 68.9 | 31.7 | 32.9 |
| BF-TFD | 33.7 | 32.2 | 29.3 | 31.9 | 58.1 | 56.2 |

## A.4   REAL PERTURBATIONS

Table 15: List of video and audio perturbation types with descriptions.

| Category | Perturbation Type | Description |
|---|---|---|
| **Video Perturbations** | Gaussian Blur | Applies Gaussian smoothing to simulate out-of-focus capture. |
| | Salt and Pepper Noise | Random white and black pixel noise, mimicking sensor errors. |
| | Low Bitrate Compression | Blocky, artifact-heavy images due to compression. |
| | Gaussian Noise | Electronic sensor noise typical in low-light conditions. |
| | Poisson Noise (Shot Noise) | Noise from photon-limited imaging environments. |
| | Speckle Noise | Multiplicative noise creating granular interference effects. |
| | Color Quantization | Banding effects from limited color palettes. |
| | Random Brightness | Simulates variations in exposure and lighting. |
| | Motion Blur | Imitates camera or object motion during capture. |
| | Rolling Shutter | Distortion effects due to CMOS sensor movements. |
| | Camera Shake | Minor frame shifts from handheld camera vibrations. |
| | Lens Distortion | Optical distortions like barrel or pincushion effects. |
| | Vignetting | Darkening of image edges typical of certain lenses. |
| | Exposure Variation | Adjusts brightness and contrast, simulating exposure issues. |
| | Chromatic Aberration | Color channel shifts causing fringing effects. |
| **Audio Perturbations** | Compression Artifacts | Quality loss from low bitrate compression. |
| | Pitch/Loudness Distortion | Gain or frequency alterations simulating recording issues. |
| | White Noise | Constant background electronic interference noise. |
| | Time Stretch | Audio speed adjustments without pitch change. |
| | Reverberation | Echo and reverb modeling room acoustics. |
| | Ambient Noise | Background environmental sounds added. |
| | Clipping | Distortion from exceeding audio amplitude limits. |
| | Frequency Filter | Filtering effects simulating transmission equipment variations. |
| | Doppler Effect | Pitch modulation due to relative motion. |
| | Interference | Static-like bursts mimicking external disturbances. |
| | Room Impulse Response | Complex echo patterns modeling specific environments. |

## A.5 AUGMENTATION EXAMPLES

In Table 16, we provide examples of augmentations achieved through the manipulation rules previously outlined in Section 3.2.1.

Table 16: Examples of augmentations achieved through the different transcript manipulation rules, showing the original (orig) and augmented (aug) transcriptions.

| Original Transcription | Original Word | Inserted Word | Operation | Example |
|---|---|---|---|---|
| CSW | EN | EN | Change meaning only (keep English) | Edit: Telephone → Radio
[orig] اتشغلت الهانم في الـ **Telephone**
(The lady got busy on the **telephone**)
[aug] اتشغلت الهانم في الـ **Radio**
(The lady got busy with the **radio**) |
| CSW | AR | AR | Change meaning only (keep Arabic variant) | Edit: منتشرة (MSA) → محدودة (MSA)
[orig] الـ Mirroring أصبحت أداة **منتشرة** جداً
(Mirorring has become a **popular** tool)
[aug] الـ Mirroring أصبحت أداة **محدودة** جداً
(Mirorring has become a **limited** tool) |
| CSW | AR | AR | Change meaning + change Arabic variant | Edit: بشكر (MSA) → بكره (Dialectal Arabic)
[orig] **بشكر** كل ال sponsors اللي موجودين
(I **thank** all the present sponsors)
[aug] **بكره** كل ال sponsors اللي موجودين
(I **hate** all the present sponsors) |
| Arabic | AR | AR | Change meaning only (keep Arabic variant) | Edit: سعيد (MSA) → حزين (MSA)
[orig] وهيكون هذا الشخص راضي و**سعيد**
(And this person will be content and **happy**)
[aug] وهيكون هذا الشخص راضي و**حزين**
(And this person will be content and **sad**) |
| Arabic | AR | AR | Change meaning + change Arabic variant | Edit: جوهري (MSA) → تافه (Dialectal Arabic)
[orig] كانت تشترك بعمل أساسي **جوهري**
(She was involved in a core and **essential** task)
[aug] كانت تشترك بعمل أساسي **تافه**
(She was involved in a core and **non-essential** task) |
| Arabic | EN | EN | Change meaning + change language to English | Edit: الناس → friends
[orig] أنا بروح قابل **الناس**
(I go meet **people**)
[aug] أنا بروح قابل **friends**
(I go meet **friends**) |

## A.6 PROMPT FOR TEXT PERTURBATION

---

**Prompt for Fake Transcript Generation.**

```
###SYSTEM MESSAGE###
You are a controlled text-perturbation bot.
Here is the transcript of an audio.
Please use the provided operations to modify
the transcript to change its sentiment.
The operation can be one of `delete`,
`insert` and `replace`.
Please priority modify adjectives and adverbs.
------------------CHANGE-MODES------------------
• meaning_only
     – Change the *meaning* of one word.
     – Keep the same language/script and dialect.
• dialect_only
     – Swap a word for a dialectal equivalent of *identical meaning*.
     – Example: <syArT> → <`rbyT> (Gulf dialect, same meaning).
• dialect_plus_meaning
     – Change *both* dialect *and* meaning in a single word.
     – Example: <jmyl> (msa, 'nice') → <wH$> (Egyptian, 'awful').
• meaning_plus_translation
     – In Arabic-only sentences, pick a word that
     is **commonly code-switched
to English** in everyday speech (e.g., <mwbayl>, <syArT>, ).
     – Translate that word to English and change the
     meaning simultaneously.
        Example: <syArT> ('car') → bike.
------------------CSW MULTI-OP LOGIC------------------
If language == 'csw':
  num = 1  → edit exactly one token matching target_token_script.
  num = 2  → edit 1 English + 1 Arabic token.
  num = 3  → edit 1 English + 2 Arabic tokens.
  ------------------OTHER RULES------------------
• Only modify tokens that are *commonly code-switched* in real speech
  (brand names, technology, everyday nouns, etc.).
• Each operation targets ONE word (delete / insert / replace).
• Number of operations for INSERT, DELETE and REPLACE
should be equal across
the data.
• If sentiment can be changed with INSERT or DELETE,
prefer it over REPLACE.
• When dialect shifts, include original_dialect and new_dialect.
• Never alter tense or add restricted content.
• Return **only** a JSON object that matches the schema.
```

Figure 6: System prompt for text-perturbation bot

## A.7 END USER LICENSE AGREEMENT (EULA FORM)

### End User License Agreement.

```
End User License Agreement
(Academic, non-commercial, not-for-profit licence)
Copyright (c) 2025 ....[AUTHORS]
All rights reserved.
The goal of the ArEnAV database is to develop new techniques, technology, and algorithms
for multimodal, code-switched deepfake detection and localization, as most of the existing research
↪   focuses on monolingual content, often overlooking the challenges of multilingual and
↪   code-switched speech, where multiple languages are mixed within the same discourse. The licensors
↪   are involved in an ongoing effort to strengthen detection algorithms against highly realistic
↪   deepfakes. The dataset is meant to aid research efforts in the general area of developing,
↪   testing and evaluating algorithms for multilingual code-switched deepfake detection and
↪   localization.

To receive a copy of the dataset, the requester must agree to observe the conditions listed Below.

The goal of the ArEnAV database is to develop new techniques, technology, and
algorithms for predicting and locating (with timestamps) where a video has been
manipulated, particularly when it has Arabic-English code-switching. Use is permitted of the
↪   databases and annotations above in source and binary form, provided that the following
conditions are met:
• The database is provided under the terms of this license strictly for academic,
non-commercial, not-for-profit purposes.
• Requestor needs to supply their university IRB application ID, and every project member must use an
↪   individual account, safeguarding traceability and preventing misuse. Attach the IRB approval in
↪   the email along with the signed EULA form.
• Redistribution, republishing, or dissemination in any form, source or binary, is not permitted
↪   without prior written approval by the licensors. Linking to the webpage of the database [WEB LINK
↪   HERE] is permitted.
• The names of the licensors may not be used to endorse or promote products
derived from this software without specific prior written permission.
• The licensors reserve the right to modify the data/license at any point.
Modification of the database by licensees is not permitted.
• In no case should the still frames or videos be used in any way that could cause the original
↪   subject embarrassment or mental anguish.
• You understand that the ArEnAV dataset is a deepfake dataset generated based
on VisPer ([2406.00038] ViSpeR: Multilingual Audio-Visual Speech Recognition)
dataset's Arabic Train subset. You also agree to all agreements of the VisPer
dataset.
• The authors of the dataset make no representations or warranties regarding the
dataset, including but not limited to warranties of non-infringement or fitness for a particular
↪   purpose.
• You accept full responsibility for your use of the dataset and shall defend and
indemnify the Authors of ArEnEV, against any and all claims arising from your use of the dataset,
↪   including but not limited to your use of any copies of copyrighted images that you may create
↪   from the dataset.
• Any publications arising from the use of this software, including but not limited to academic
↪   journal and conference publications, technical reports and manuals, must cite the following
↪   works:
[CITATION]

THE DATABASE IS PROVIDED BY THE AUTHORS "AS IS" AND ANY EXPRESS OR IMPLIED WARRANTIES, INCLUDING, BUT
↪   NOT LIMITED TO, THE IMPLIED WARRANTIES OF MERCHANTABILITY AND FITNESS FOR A PARTICULAR PURPOSE
↪   ARE DISCLAIMED. IN NO EVENT SHALL THE COPYRIGHT HOLDER OR CONTRIBUTORS BE LIABLE FOR ANY DIRECT,
↪   INDIRECT, INCIDENTAL, SPECIAL, EXEMPLARY, OR CONSEQUENTIAL DAMAGES (INCLUDING, BUT NOT LIMITED TO,
↪   PROCUREMENT OF SUBSTITUTE GOODS OR SERVICES; LOSS OF USE, DATA, OR
PROFITS; OR BUSINESS INTERRUPTION) HOWEVER CAUSED AND ON ANY THEORY OF LIABILITY, WHETHER IN CONTRACT,
↪   STRICT LIABILITY, OR TORT (INCLUDING NEGLIGENCE OR OTHERWISE) ARISING IN ANY WAY OUT OF THE USE
↪   OF THIS DATABASE, EVEN IF ADVISED OF THE POSSIBILITY OF SUCH DAMAGE. THE PROVIDER OF THE DATABASE
↪   MAKES NO REPRESENTATIONS AND EXTENDS NO WARRANTIES OF ANY KIND, EITHER EXPRESSED OR IMPLIED.
↪   THERE ARE NO EXPRESS OR IMPLIED WARRANTIES THAT THE USE OF THE MATERIAL WILL NOT INFRINGE ANY
↪   PATENT, COPYRIGHT, TRADEMARK, OR OTHER PROPRIETARY RIGHTS. If you have read and understood the
↪   user agreement and will comply with it.

Signed
_______________________________
Print Name
_______________________________
Institution Name
_______________________________
Date
_______________________________
Addition Researcher 1
_______________________________
Addition Researcher 2
_______________________________
```

Figure 7: End User License Agreement for accessing ArEnAV.

## A.8 LLM USAGE

Along with the use of Large Language Models (LLMs) as described in our Data-Creation process, we made limited use of LLMs to enhance the clarity and readability of the text. They were not involved in the conception of ideas, the design of experiments, analysis, or the production of results.

