# OpenReview forum: "Tell me Habibi, is it Real or Fake?"
_ICLR.cc/2026/Conference — ICLR 2026 Poster_

### Official Review · Reviewer_Wf9J · 2025-10-29

**Soundness:** 3
**Presentation:** 2
**Contribution:** 3
**Rating:** 8
**Confidence:** 4

**Summary:**

This work introduces ArEnAV, a large-scale Arabic-English audio-visual deepfake dataset, designed to address a critical gap in current research: the challenge of detecting deepfakes within multilingual and code-switched content. The work's core contributions are twofold: 1) the ArEnAV dataset itself, the first large-scale benchmark featuring intra-utterance code-switching and dialectal variations; and 2) a novel data generation pipeline that leverages Large Language Models for content manipulation and integrates SOTA TTS and lip-sync models to generate high-fidelity forgeries. Comprehensive benchmark results highlight the dataset's challenging nature, demonstrating a significant performance drop in state-of-the-art models. These findings validate the limitations of current detectors that are predominantly trained on monolingual data.

**Strengths:**

1. This work successfully tackles an important and overlooked problem: detecting audio-visual deepfakes in code-switched (CSW) speech. This is a major step towards building deepfake detectors that work in the real world.
2. This work proposes ArEnAV, a new large-scale dataset for this task. The pipeline used to create the data is novel and combines several SOTA models, providing a valuable new resource for the research community.

**Weaknesses:**

1. The primary evaluation metric (AP@IoU=0.5) may be poorly suited for the dataset's extremely short, single-word forgeries.
2. The "TTS and insert" audio generation method can create unnatural splice artifacts, which may affect the dataset's validity. These artifacts could allow models to detect forgeries using simple audio errors rather than the intended code-switching cues, thus misrepresenting the true nature of the detection challenge.

**Questions:**

1. The paper notes that the LLM did not always change the meaning in the "meaning + translation" mode. Did you try to fix this? If not, how do you know these samples did not lower the dataset's overall difficulty and affect your final conclusions?
2. Could you provide the average, minimum, and maximum duration of the fake words in your dataset? This information is very important. Without it, we cannot be sure if the poor model performance is because the task is hard, or because your evaluation metric (AP@IoU=0.5) is simply too strict for such short fake clips.

---

> ### Author Response · Authors · 2025-11-21
>
> We thank the reviewer for their constructive feedback. Below, we answer all the questions and weaknesses mentioned:
>
> ---
>
> ***q1) The paper notes that the LLM did not always change the meaning in the "meaning + translation" mode…***
>
> We acknowledge that in some cases, the “meaning + translation” mode produced sentences that were semantically similar to the originals. This can be improved by reprocessing the LLM outputs and filtering based on entailment or cosine distance to enforce greater semantic variation. However, this does not affect our main conclusions. The detection task in ArEnAV focuses on identifying audio-visual inconsistencies, not the extent of semantic change.
>
> Only a small fraction of the data (less than 8,000 out of 280,000 samples, and about 2,000 out of 78k in the test set) (Figure 3a) has an entailment quality mean > 0.8. To confirm this, we compared the best-performing models across these subsets. As shown in Table 3, the performance remains consistent, indicating that these few cases do not lower the dataset’s overall difficulty.
>
> Table 3. Performance comparison of BA-TFD and BA-TFD+ on ArEnAV test set (split into subset with entailment over and below 0.8) (AUC scores).
>
> | Model | Overall | Entailment < 0.8 | Entailment > 0.8 |
> | --- | --- | --- | --- |
> | **BA-TFD+** | 79.97 | 79.96 | 79.98 |
> | **BA-TFD** | 75.91 | 75.92 | 75.89 |
>
> ---
>
> ***q2) Could you provide the average, minimum, and maximum duration of the fake words in your dataset…***
>
> The fake word (forged segment) durations in **ArEnAV** are tightly distributed around short, word-level edits, as expected for intra-utterance code-switching manipulations. Across all splits, the **average fake segment duration is 0.696 seconds**, with a **minimum of 0.02 seconds** and a **maximum of 6.16 seconds**. The median duration is **0.625 seconds**, and **90% of all fake spans are shorter than 1.12 seconds**. The corresponding videos have an average total length of **5.97 seconds**, meaning that each fake region accounts for roughly **12.1% of the total clip duration**.
>
> In contrast, the AV-1M dataset reports an average fake segment length of only **0.326 seconds**, an average fake proportion of **3.7%**, and a mean video duration of **9.07 seconds**. Thus, fake spans in ArEnAV are approximately **2.1× longer** and occupy a **threefold larger proportion of the video**. These statistics confirm that the observed performance drop is not an artifact of overly short fake spans or an overly strict AP@IoU=0.5 metric, but instead reflects the real difficulty of identifying precise, linguistically meaningful intra-utterance manipulations in multilingual speech.
>
> We will add these statistics to the main paper.
>
> Table 4. Forged segment duration statistics.
>
> | Metric | ArEnAV | AV-1M |
> | --- | --- | --- |
> | Mean fake segment duration (s) | 0.696 | 0.326 |
> | Median fake segment duration (s) | 0.625 | — |
> | Min fake segment duration (s) | 0.02 | — |
> | Max fake segment duration (s) | 6.16 | — |
> | 90th percentile fake duration (s) | 1.12 | — |
> | Mean video duration (s) | 5.97 | 9.07 |
> | Mean fake proportion of video (%) | 12.1 | 3.7 |
> | Relative length vs. AV-1M (×) | 2.1× | 1.0× |
> | Relative fake proportion vs. AV-1M (×) | 3.0× | 1.0× |
>
> ---
>
> ***w1) The "TTS and insert" audio generation method can create unnatural splice artifacts, …***
>
> We mitigate potential splice artifacts through loudness normalization and denoising during audio insertion (lines 275, 260), ensuring smooth transitions and natural continuity. Moreover, both human evaluators (Table 5) and finetuned detection models (Tables 8, 9, 10) perform poorly on the dataset, confirming that the challenge arises from genuine multimodal–multilingual complexity rather than simple audio artifacts. Figure 4 (main paper) further supports this: qualitative output examples from BA-TFD+ on ArEnAV fake videos show that the predictions are not concentrated at the splices or joining regions.

---

> ### Author Response · Authors · 2025-11-28
>
> Thank you again for your thoughtful feedback. We have thoroughly replied to each of the concerns raised in your earlier comments. As the author–reviewer discussion phase remains open, we would greatly appreciate any follow-up questions or clarifications you may have. Your additional input will help ensure a fair and thorough assessment of our work.

---

### Official Review · Reviewer_8ZWD · 2025-11-02

**Soundness:** 2
**Presentation:** 2
**Contribution:** 2
**Rating:** 4
**Confidence:** 3

**Summary:**

The authors in the paper point out the lack of a large-scale multilingual dataset that includes both English and Arabic, particularly one featuring code-switching between the two languages. They emphasize that code-switching is highly prevalent in daily speech across the Arabic world. To address this gap, they introduce the first large-scale Arabic–English audio-visual deepfake dataset, which features intra-utterance code-switching, dialectal variation, and monolingual Arabic content. It contains 387k videos and over 765 hours of real and fake videos generated by the up-to-date SOTA model.

**Strengths:**

- It will be released as an open-source, large-scale bilingual dataset. Given that Arabic is spoken by hundreds of millions of people all over the world, the dataset holds significant importance.

- The data generation pipeline was clearly described in the paper, enabling easy reproducibility.

- The quality of the generated fake data was comparable with the well known dataset AV-Deepfake1M, as evaluated by standard metrics.

**Weaknesses:**

1. Insufficient Direct Experimental Evidence for the "Code-Switching" Contribution

The paper's central contribution is its focus on a multilingual and code-switching (CSW) dataset. However, the experimental results presented in Tables 8, 9, and 10a do not directly prove that this code-switching characteristic is the key factor driving the dataset's difficulty. The authors demonstrate that existing models perform poorly and attribute this failure to the novelty of CSW.
Specifically, the zero-shot performance drop in the temporal localization task (Table 8) is a weak argument. To my knowledge, models like BA-TFD were not designed for zero-shot generalization to new domains, languages, and generation methods. Their failure on this cross-domain dataset is expected and does not isolate the impact of code-switching itself.

2. Absence of "Close-Inspection" and Qualitative Analysis

Connecting to the first point, the paper lacks a "close-inspection" of why the models fail. The analysis relies heavily on aggregate metrics (AUC, AP), which only show that models fail, not the cause. To sophisticate the claim that CSW and multilingualism are the core challenges, a deeper analysis of the fine-tuned models is necessary.
The authors could provide qualitative, frame-level examples of incorrect predictions,
But with current state, paper cannot clearly answer the question below.
Do the models consistently fail at or around the code-switched regions? Or are the errors more correlated with the high-quality diffusion-based lip-sync, specific audio perturbations, or other artifacts?
It is difficult to disentangle the source of the dataset's complexity. It is unclear whether this dataset is challenging because of its CSW properties or because of the careful, high-quality generation techniques and perturbations the authors have induced.



3. Use of Outdated Benchmark Models

The choice of models for benchmarking (Meso4, MesoInception4, Xception) is a notable weakness. These are significantly outdated models that are widely known to overfit specific dataset artifacts and lack generalization capabilities.
While BA-TFD is included for the temporal task, the overall detection model suite is quite naive. Relying on these older architectures makes it difficult to assess if the reported performance drop is due to the dataset's genuine complexity or simply the known limitations of these models. The benchmark would be far more compelling if more recent, state-of-the-art detection models with demonstrated generalization capabilities and evaluated.

Minor:
The result for LAA-Net listed under the DFDC dataset in Table 10b is incorrect.
That specific performance metric was obtained on the DFDC-P (DFDC Pre-processed) dataset, as noted in the original citation by the authors.
(See: https://github.com/10Ring/LAA-Net/issues/3)



4. The protocol of the data generation pipeline is similar to the approach used in generating fake data for AV-Deepfake1M. No novel architecture for fake data generation was introduced.

5. The authors did not conduct an intra-dataset evaluation to assess how code-switching affects the accuracy of temporal localization or deepfake detection models within the same domain. Instead, they evaluated models which were trained on a dataset containing a small number of Arabic videos (presumably without code-switching) and then tested on the proposed ArEnAV dataset. However, this is across domain evaluation, so the drop in accuracy is expected.

6. The dataset also shows limited instruction following in code-switching scenarios; the authors relied on the GPT model to generate code-switching transcripts, making real and fake transcripts too similar and not always changing their meaning.


7. The motivation is sound, but the authors have not considered the models such as Diff2Lip and LatentSync, are trained on English speaking videos, and the natural lip-sync to real-life Arabic speaking phonemes are different than English, including the speed of which Arabic is spoken. The authors should specify the limitations and rationale of using this.

8.	In Table 10 training protocol is not described. As fake part can be anywhere in the video, so in this experiment the split of video clips, is not defined, also it is not written how much length of video is used to test? Similarly, the details of the experiment should be disclosed for fair comparison.

9.	The authors used 4 commercial audio generation methods, while for visual manipulations the authors used only 2, which are also not commercial. Visual manipulation is harder to detect; hence diversity can help more generalizable deepfake detection.

10.	3.2.2 Audio generation pipeline is very messy, hard to comprehend, a flow chart could help understand this better.

11. Figure 1 text is very small can be improved, also the fonts used for Arabic can be improved for better readability.

12. 	Although language overlap is addressed, the identity split is not defined.

13.	Meta data information is not provided.

14. The dataset contains imbalance samples, which may introduce bias while training.

**Questions:**

1. Why was there no intra-dataset evaluation for the necessity of code-switching manipulation?

2. Could you provide a few qualitative examples (e.g., video frames or audio spectrograms) of your fine-tuned models' failure cases? We need to see where the models are failing. Are the incorrect predictions concentrated around the temporal boundaries of the code-switch? Or are they failing due to other artifacts (like the lip-sync) that are unrelated to the CSW

3. Table 10a shows comparison with the proposed partial fake dataset with full fake dataset, is it fair comparison?

4. Did the author make the dataset identity disjoint? Also does the dataset contain real and fake pair information in metadata?

---

> ### Author Response · Authors · 2025-11-21
>
> We thank the reviewer for their constructive feedback. Below, we answer all the questions and weaknesses mentioned by the reviewer.
>
> ---
>
> ***q1, w1, w5) Why was there no intra-dataset evaluation…***
>
> We would like to clarify that Tables 8 and 9 present intra-dataset evaluations. The models reported under the “video-level” and “frame-level” sections were trained on the ArEnAV training and validation splits and evaluated on the held-out test split from ArEnAV. Only the entries marked as “zero-shot” correspond to models evaluated without fine-tuning and are included only for comparison. For cross-dataset comparison, refer to Table 10 (lines 428).
>
> It is not entirely clear what specific form of intra-dataset comparison the reviewer is referring to. We would be happy to clarify our approach or include the requested analysis if the reviewer could elaborate further or provide some reference.
>
> ---
>
> ***q2, w2) Qualitative failure examples…***
>
> We provide a qualitative analysis of model failures in Figures 4 and 5 (main paper), which present representative outputs from BA-TFD+ after fine-tuning on ArEnAV. Figure 4 illustrates model predictions on fake samples, showing four representative cases comparing predicted fake segments with ground-truth regions. In some cases, the model correctly identifies the manipulated class with good temporal overlap, while in others, the predictions are either misaligned or incorrect. Importantly, these outputs demonstrate that the model’s behavior is not influenced by any boundary-level artifacts or splicing cues introduced during data generation.
>
> Figure 5 shows the model’s predictions on real code-switched videos. Here, the model frequently misclassifies real videos as fake, primarily due to the presence of code-switching between Arabic and English. The high predicted fake scores at these regions indicate that the model confuses natural linguistic transitions with synthetic inconsistencies. Together, these qualitative results confirm that the challenge in ArEnAV arises from the intrinsic complexity of code-switching rather than from generation artifacts.
>
> ---
>
> ***q3, w8) Clarification about  table 10a***
>
> We would like to clarify that Table 10(a) presents a fair and direct comparison. The BA-TFD and BA-TFD+ models were each trained on AV-1M and evaluated on respective test sets of LAV-DF, AV-1M, and ArEnAV. LAV-DF and AV-1M are English-only partial deepfake datasets that use similar generation methods, making them appropriate baselines.
>
> The results show that although models generalize from AV-1M to LAV-DF (English-only partial deepfake datasets), both models experience a significant drop in performance on ArEnAV. This demonstrates that the challenge arises from the code-switching and multilingual variability in ArEnAV, not from differences in dataset setup or training protocol.
>
> ---
>
> ***q4, w12) Did the author make the dataset identity disjoint? Also does the dataset contain real and fake pair information in metadata?***
>
> Identity disjoint:
>
> Thank you for raising this point. On diving deeper, we conducted a face analysis using InsightFace and DBSCAN to better understand potential identity overlaps across dataset splits. While each split (train, validation, and test) already contains independent YouTube video IDs, we found a 41.7% identity match between the combined train+val and test sets at a similarity threshold of 0.7.
>
> To verify whether this overlap influences model performance, we evaluated BA-TFD and BA-TFD+ separately on overlapping and non-overlapping subsets. As shown in Table 1, the results remain consistent across both groups, indicating that identity overlap does not affect the models’ behavior. We will release a separate identity-disjoint test set in the final version.
>
> Table 2. Performance of BA-TFD and BA-TFD+ models on ArEnAV test set. Each column shows Accuracy@0.5 and AUC, with the number of evaluated videos in parentheses.
>
> | **Model** | **Overall (78,400)** Acc | **Overall AUC** | **Non-Overlapping (45,512)** Acc | **Non-Overlapping AUC** | **Overlapping (32,888)** Acc | **Overlapping AUC** |
> | --- | --- | --- | --- | --- | --- | --- |
> | **BA-TFD+** | 27.44 | 79.97 | 27.58 | 79.96 | 27.31 | 79.99 |
> | **BA-TFD** | 44.31 | 75.91 | 44.35 | 75.96 | 44.27 | 75.85 |
>
> Metadata:
>
> The metadata already includes source video references for every fake clip to enable full traceability.

---

> > ### Author Response · Authors · 2025-11-21
> >
> > ***w3) Use of Outdated Benchmark Models…***
> >
> > While we included earlier baselines (Meso4, MesoInception4, Xception, CapsuleNet-v2) for continuity with prior datasets, we also evaluated recent state-of-the-art detection models, as reported in Table 10(b). These include LipForensics (CVPR 2021), M2TR (ICMR 2022), LAA-Net (CVPR 2024), and ForensicsAdaptor (CVPR 2025).
> > Even these modern architectures, which have demonstrated strong generalization across benchmarks, show near-random performance on ArEnAV. This confirms that the difficulty arises from the dataset’s code-switching and multilingual complexity, rather than from model limitations.
> >
> > *w3) Minor:* The result for LAA-Net listed under the DFDC dataset in Table 10b is incorrect…
> >
> > Thank you for pointing it out. We will fix this in the camera-ready version.
> >
> > ***w4)…No novel architecture for fake data generation was introduced.***
> >
> > Our work introduces clear technical novelty through a controllable, word-level intra-utterance code-switching (CSW) pipeline that manipulates transcripts before audio and visual synthesis. This allows linguistically precise mixing of Arabic and English within a single sentence. Each edited transcript is validated through ASR alignment and multimodal quality checks, integrating multi-speaker TTS and diffusion-based lip-sync models in a modular and reproducible setup.
> >
> > ***w6) The dataset also shows limited instruction following in code-switching scenarios…***
> >
> > We acknowledge that in some cases, the “meaning + translation” mode produced sentences that were semantically similar to the originals. This can be improved by reprocessing the LLM outputs and filtering based on entailment or cosine distance to enforce greater semantic variation. However, this does not affect our main conclusions. The detection task in ArEnAV focuses on identifying audio-visual inconsistencies, not the extent of semantic change.
> >
> > Only a small fraction of the data (less than 8,000 out of 280,000 samples, and about 2,000 out of 78k in the test set) (Figure 3a) has an entailment quality mean > 0.8. To confirm this, we compared the best-performing models across these subsets. As shown in Table 3 (in response to the reviewer **Wf9J**), the performance remains consistent, indicating that these few cases do not lower the dataset’s overall difficulty.
> >
> > ***w7) …the authors have not considered that models such as Diff2Lip and LatentSync are trained on English-speaking videos…***
> >
> > Both human evaluation (Table 6) and video-only model performance (Table 8,9) indicate that these models produce lip-sync quality that is realistic enough to challenge detectors, suggesting they perform adequately for our purpose.
> > We will note this explicitly in the Limitations section. While current models work reasonably well, our pipeline is fully modular, and any future multilingual or Arabic-specific lip-sync model can be easily integrated to further improve lip movement accuracy and naturalness.
> >
> > ***w9) The authors used 4 commercial audio generation methods, while for visual manipulations the authors used only 2, which are also not commercial…***
> >
> > Adding more visual manipulation techniques could increase diversity and improve generalization. Since our pipeline is fully modular, additional lip-sync or face reenactment models can be easily integrated in future versions to further enhance the visual variability of the dataset.
> >
> > ***w10, w11) 3.2.2 Audio generation pipeline is very messy… Figure 1 text is very small ..***
> >
> > We will make these changes in our camera-ready version. Thank you.
> >
> > ***w13) Metadata information is not provided.***
> >
> > We will provide all the metadata information along with our dataset release, which will contain the source video, audio/video models used, ground truth transcripts, and edited transcripts.
> >
> > ***w14) The dataset contains imbalanced samples, which may introduce bias while training.***
> >
> > All model training was performed with sub-sampling to maintain class balance between real and fake clips. This ensured that learning and evaluation were not biased by dataset imbalance. The consistent performance trends across models confirm that the observed difficulty is due to the complexity of code-switched content, not data imbalance. We will clarify this in the Implementation section for completeness and upload the PDF file with the changes shortly.

---

> ### Author Response · Authors · 2025-11-28
>
> Thank you again for your thoughtful feedback. We have thoroughly replied to each of the concerns raised in your earlier comments. As the author–reviewer discussion phase remains open, we would greatly appreciate any follow-up questions or clarifications you may have. Your additional input will help ensure a fair and thorough assessment of our work.

---

### Official Review · Reviewer_W12a · 2025-11-03

**Soundness:** 3
**Presentation:** 3
**Contribution:** 2
**Rating:** 4
**Confidence:** 3

**Summary:**

This paper introduces ArEnAV, the first large-scale Arabic–English code-switching (CSW) audio-visual deepfake dataset, consisting of 387K videos (765+ hours) with intra-utterance CSW, dialect variation, and multiple manipulation types. The authors propose a multi-stage generation pipeline combining GPT-4.1-mini-based transcript manipulation, four TTS systems, and two diffusion-based lip-sync models. The paper benchmarks several state-of-the-art deepfake detection and localization models and shows a drastic performance drop when evaluated on ArEnAV, demonstrating the dataset’s difficulty and relevance. A user study further confirms that humans also struggle to detect these deepfakes (≈60% accuracy). The dataset and code are promised to be released.

**Strengths:**

1.	Existing deepfake datasets are monolingual or multilingual but lack intra-utterance code-switching. The paper clearly identifies this gap and addresses it convincingly.
2.	Large-scale, well-engineered dataset. 387K videos, 4 TTS + 2 lip-sync models, stratified splits, strong statistics, and detailed generation pipeline. The dataset is significantly larger and more diverse than prior multilingual datasets. The authors show that state-of-the-art models (e.g., BA-TFD, LipForensics, Capsule-v2) perform poorly, even close to random guessing on some settings, demonstrating real difficulty.
3.	Human performance ≈60% accuracy, poor localization ability → confirms that deepfakes in code-switching settings are hard even for humans, not only for models.

**Weaknesses:**

1.	The paper does not propose any new detection model or algorithm. I feel the work as “engineering + dataset release” rather than a scientific advance.
2.	Heavy reliance on closed-source models (GPT-4.1, Whisper, TTS-1, etc.). Reproducibility is partially limited. If OpenAI APIs change, future users may not be able to regenerate the dataset. This may be flagged in the reproducibility checklist.
3.	Although CSW is the main motivation, the paper lacks deeper linguistic validation:
⦁	Is the LLM-generated code-switching natural vs synthetic?
⦁	How does the CSW distribution compare to real-world corpora?
⦁	Does GPT-4.1 make linguistically plausible switching decisions?
4.	Real vs fake imbalance is acknowledged but not studied. No experiments showing how class imbalance affects model learning. Meanwhile, generalization to other multilingual settings not demonstrated.

**Questions:**

1.	Could the authors clearly articulate which parts of the pipeline are technically novel, and whether it is reusable beyond this dataset?
2.	If these APIs change or become unavailable, can the dataset still be regenerated?
3.	Did the authors run any linguistic validation (human or automatic) to ensure the generated CSW resembles real corpora like ZAEBUC or ArzEn?
4.	Can the authors provide quantitative evidence on how different TTS/lip-sync components affect detectability or quality?
5.	How does this imbalance affect model training? Did the authors experiment with balancing, reweighting, or sub-sampling?
6.	Can the authors comment on whether the pipeline could scale to other CSW settings (e.g., Hindi-English, Spanish-English)?

**Details Of Ethics Concerns:**

The dataset contains real public YouTube videos featuring identifiable individuals, but the paper does not state whether explicit consent was obtained, relying instead on “fair use” and a gated EULA. It is unclear whether this satisfies GDPR/international privacy rules, especially given redistribution of face and voice data.
Additionally, the dataset enables high-quality multimodal deepfake generation, which carries dual-use risks. The paper does not discuss safeguards beyond access control, nor whether the authors conducted a formal risk–benefit assessment.
Finally, the human study involves 19 participants but no explicit IRB approval number is provided. Clarification on ethical handling of human evaluation data is requested.

---

> ### Author Response · Authors · 2025-11-21
>
> We thank the reviewer for their constructive feedback. Below, we answer all the questions and weaknesses mentioned by the reviewer.
>
> ***q1, q6) …which parts of the pipeline are technically novel, and whether it is reusable beyond this dataset?***
>
> ***Novelty***
>
> Our pipeline introduces a technically novel approach for controllable, word-level intra-utterance code-switching (CSW). It manipulates transcript text directly before audio and video synthesis, allowing fine-grained and linguistically accurate mixing of Arabic and English within the same sentence (section 3.2.1). This is the first large-scale system to achieve such controlled CSW in audio-visual deepfake generation (line 87-88). Each edited transcript is validated through automatic speech recognition (ASR) alignment and strict synchronization checks so that only perfectly aligned audio-text pairs are retained (section 3.2.2). The pipeline also combines multi-speaker text-to-speech (TTS) and diffusion-based lip-sync models, together with detailed quality verification using semantic (NLI), acoustic (SECS, SNR, FAD), and visual (PSNR, SSIM, FID) metrics (lines 278-289, 298-307).
>
> ***Reusability***
>
> The pipeline is fully language-pair agnostic and can be scaled to other code-switching settings such as Hindi–English or Spanish–English. The transcript manipulation stage only requires bilingual text prompts, while the ASR and alignment components rely on multilingual models like Whisper and wav2vec2. All TTS and lip-sync modules are modular, allowing direct substitution with language-specific models.
>
> In practice, the current components already support a wide range of languages: Whisper (54), GPT-4.1-mini (50+), GPT-TTS (9), Fairseq (~1000), XTTS-v2 (17), and wav2vec2 (39). Thus, the pipeline can be used out of the box for most bilingual or multilingual language pairs without major modification.
>
> ---
>
> ***q2, w2) If these APIs change or become unavailable...***
>
> Since we plan to make the dataset (along with all the intermediate metadata) publicly available, future users will not need to regenerate the ArEnAV dataset from scratch. However, for generating code-switched data in other language pairs, our pipeline is completely modular, and any model can be substituted for another without affecting the rest of the pipeline. Besides, all the stages like TTS, lip-sync, and audio-visual rendering are based on **open-source models** such as XTTS-v2, OpenVoice-v2, Fairseq-Arabic, Diff2Lip, and LatentSync. The only closed component we used was GPT-4.1-mini for the transcript-editing stage, and we will release all the **prompts, seeds, edited transcripts, and forced alignment manifests**.
>
> ---
>
> ***q3, w3) Did the authors run any linguistic validation…***
>
> Yes. We conducted automatic linguistic validation using **perplexity analysis** (Figure 3b) to verify that the generated CSW text maintains natural fluency and surface structure comparable to real bilingual speech. Specifically, we compared the per-utterance perplexity of our generated CSW transcripts and the real transcripts against the unaltered transcripts from the **SDAIANCAI dataset**, which contains English–Arabic code-switching text and serves as our control dataset (yellow bar in Figure 3b). Using large Arabic–English language models (Jais-3B and Qwen-2.5-7B), we found that distributions of real and fake transcripts are nearly identical, indicating that the code-switched text remains linguistically coherent and fluent. This confirms that our edits preserve natural language patterns while introducing realistic bilingual alternation.
>
> Further, our user study had 15 native Arabic speakers (line-318), who were confused while identifying the manipulated videos, indicating that the code-switching was, in fact, linguistically plausible and close to natural.
>
> ---

---

> > ### Author Response · Authors · 2025-11-21
> >
> > ***q4) Can the authors provide quantitative evidence on how different TTS/lip-sync components affect detectability or quality?***
> >
> > Table 1 reports model accuracy across all TTS and lip-sync configurations (finetuned on ArEnAV). For audio-only detection, XLSR-Mamba achieves the highest accuracy with Fairseq-OpenVoice (95.8%), indicating that lower-quality TTS outputs are more easily detected. Higher-fidelity systems such as XTTS-OpenVoice and XTTS-v2 reduce detectability, suggesting improved synthesis quality.
> >
> > For audio-visual models (BA-TFD and BA-TFD+), both Diff2Lip and LatentSync yield similar accuracy levels (around 31–33%), confirming stable and high-quality visual performance. Overall, the results show that better generation quality leads to lower detectability, aligning with our goal of creating challenging, realistic benchmarks.
> >
> > Table 1. Accuracy of different Audio and Video generation models used in ArEnAV.
> >
> > | Model Type | Fairseq-OpenVoice | GPT-TTS | XTTS-OpenVoice | XTTS-v2 | Diff2Lip | LatentSync |
> > | --- | --- | --- | --- | --- | --- | --- |
> > | XLSR-Mamba | 95.8 | 78.8 | 19.4 | 77.9 | 38.6 | 42.8 |
> > | BA-TFD+ | 69.5 | 58.9 | 7.7 | 68.9 | 31.7 | 32.9 |
> > | BF-TFD | 33.7 | 32.2 | 29.3 | 31.9 | 58.1 | 56.2 |
> >
> > ---
> >
> > ***q5)w4) How does this imbalance affect model training…***
> >
> > All model training was performed with sub-sampling to maintain class balance between real and fake clips. This ensured that learning and evaluation were not biased by dataset imbalance. The consistent performance trends across models confirm that the observed difficulty is due to the complexity of code-switched content, not data imbalance. We will clarify this in the Implementation section for completeness and upload the PDF file with the changes shortly.
> >
> > ---
> >
> > ***w1) …I feel the work as “engineering + dataset release” rather than a scientific advance.***
> >
> > Our work introduces clear technical novelty through a controllable, word-level intra-utterance code-switching (CSW) pipeline that manipulates transcripts before audio and visual synthesis. This allows linguistically precise mixing of Arabic and English within a single sentence. Each edited transcript is validated through ASR alignment and multimodal quality checks, integrating multi-speaker TTS and diffusion-based lip-sync models in a modular and reproducible setup.
> > We note that several recent ICLR-accepted works, such as Illusion [1] and MMAD [2], also focus primarily on dataset creation and benchmarking. In a similar vein, our work advances deepfake detection by addressing the emerging challenge of code-switched communication. We introduce the first large-scale Arabic–English code-switched deepfake dataset and a reproducible framework for multilingual synthesis.
> >
> > [1] ILLUSION: Unveiling Truth with a Comprehensive Multi-Modal, Multi-Lingual Deepfake Dataset
> >
> > [2] MMAD: A Comprehensive Benchmark for Multimodal Large Language Models in Industrial Anomaly

---

> ### Author Response · Authors · 2025-11-28
>
> Thank you again for your thoughtful feedback. We have thoroughly replied to each of the concerns raised in your earlier comments. As the author–reviewer discussion phase remains open, we would greatly appreciate any follow-up questions or clarifications you may have. Your additional input will help ensure a fair and thorough assessment of our work.

---

> ### Author Response · Authors · 2025-12-02
> **Regarding Ethics Flag**
>
> We thank the reviewer for raising the flags about the ethics. We answer all the raised issues as follows:
>
> ### **Use of Public YouTube Videos, Consent, and GDPR/Privacy Compliance**
>
> Our dataset uses only publicly available YouTube videos and follows the same research practice established by widely used benchmarks such as LRS3-TED, VoxCeleb, AVSpeech, FakeAVCeleb, and AV-1M. As described in the Ethics Section (lines 548–551), our use of this material is restricted to non-commercial scientific research and is accessed only under a mandatory EULA (lines 552–564). This EULA forbids redistribution, identification attempts, or any harmful use and requires that requesters provide an institutional IRB application ID before access. Because we do not release biometric templates or personal metadata, and because access is controlled and tied to institutional oversight, the dataset complies with accepted fair use and legitimate interests practices in GDPR-governed research environments, as demonstrated by widely used peer-reviewed audiovisual deepfake datasets like AV-1M, AV-1M++, and FakeAVCeleb. We will clarify this basis more explicitly in the camera-ready version.
>
> ---
>
> ### **Human Study and IRB Approval**
>
> The human study adhered to university IRB procedures. All participants were adults, participation was voluntary, all responses were anonymous, and no personal data were collected (lines 586–592). The study materials included an explanatory statement and required explicit consent before participation. All participants were explicitly asked to answer through a Google form; they could stop at any time, with no repercussions. No incomplete forms were used in the final human study. All the data shown to participants was manually verified to remove any NSFW content. All of these steps ensured proper ethical handling of the evaluation data. The IRB approval number cannot be included in the current submission because it would reveal our institutional affiliation and violate double-blind review. It will be added in the camera-ready version.
>
> ---
>
> ### **Dual-Use Risks and Potential Misuse**
>
> We acknowledge that the release of any large-scale deepfake dataset raises legitimate dual-use concerns. To mitigate these risks, access to ArEnAV is controlled through a mandatory EULA (lines 552–564) that restricts the dataset to academic and non-commercial research and explicitly prohibits any use intended to deceive, impersonate, harass, or otherwise harm individuals or communities. Access requires an institutional affiliation, an IRB application ID, and individualized user accounts, which ensures traceability and prevents anonymous or uncontrolled redistribution. While the dataset includes high-quality manipulated content, its purpose is clearly defined as benchmarking and analyzing deepfake detection and localization in multilingual settings. These institutional, legal, and procedural safeguards together provide a meaningful layer of protection against misuse, while still allowing the community to benefit from an openly released, reproducible benchmark.
>
> Compared to existing deepfake datasets such as FakeAVCeleb, which requires only a Google Form submission, and AV-1M, which requires emailing a signed EULA, ArEnAV introduces significantly stronger safeguards. In particular, we require that each requester provides an IRB application ID from their host institution and obtains an individual, identifiable user account rather than group-level access. This ensures explicit institutional oversight, full user-level accountability, and a substantially higher barrier to misuse than is typical in current dataset releases. These protections, combined with our EULA restrictions and institutional responsibility clauses, offer a more robust framework to prevent harmful applications while preserving the value of the dataset as an open and reproducible research benchmark.

---

### Author Response · Authors · 2025-12-02
**Rebuttal Summary**

---

We thank all the reviewers for the detailed feedback. We have responded to each query of the reviewers separately during our rebuttal. Below is a summarised, unified response to the main concerns raised across all reviews. We have updated the paper with all the changes mentioned in our rebuttal.
PART-1/2:

---

### 1. Technical contribution and reusability of the CSW pipeline

Our main contribution is a *controllable, word-level intra-utterance code-switching (CSW) pipeline* for audio-visual deepfake generation. The pipeline operates at the transcript level before synthesis, enabling linguistically precise mixing of Arabic and English (and dialects) within a single utterance. Edited transcripts are then validated via ASR alignment and strict sync checks so that only perfectly aligned audio–text pairs are kept.

In the rebuttal, we clarified to **W12a**/**8ZWD:**

- Our pipeline design is **modular and language-agnostic**: transcript manipulation only relies on bilingual prompts, and all downstream blocks (ASR, TTS, lip-sync) can be swapped with other models.
- The models we use (Whisper, wav2vec2, GPT-4.1-mini, GPT-TTS, Fairseq, XTTS-v2, OpenVoice, Diff2Lip, LatentSync) already cover many languages, so the same pipeline can be applied to other CSW settings (e.g., Hindi–English, Spanish–English) without architectural changes.

This addresses **W12a’s** concern that the work is “only engineering + dataset release”: the pipeline itself is a reusable, technically novel mechanism for controlled, word-level CSW generation in audio-visual deepfakes.

Secondly, we will open-source all the dataset metadata (transcripts, GPT manipulations, prompts, etc.) so that the dataset can be reproduced without depending on any specific APIs or closed-source models.

---

### 2. CSW is the main source of difficulty: quantitative **and** qualitative evidence

We explicitly show, both quantitatively and qualitatively, that the poor model performance is driven by the code-switching/multilingual setting rather than weak baselines or simple artifacts.

**Quantitative: intra-dataset and cross-dataset**

- **Intra-dataset evaluation (clarifying 8ZWD).**

    Tables 8 and 9 are *intra-dataset* results: all models there are trained on ArEnAV train/val and evaluated on the held-out ArEnAV test set. Only rows explicitly marked “zero-shot” correspond to cross-dataset evaluation. We clarified this directly in our rebuttal to **8ZWD**, without changing the experiments. These tables show that even after fine-tuning on ArEnAV, temporal localization models (BA-TFD/BA-TFD+) and frame/video-level detectors still perform poorly (very low AP@0.5 and moderate AUC).

- **Cross-dataset comparisons.**

    Table 10a compares BA-TFD/BA-TFD+ trained on AV-1M and tested on LAV-DF and AV-1M (English-only partial fake datasets) versus ArEnAV. The models generalize well across LAV-DF and AV-1M but suffer a sharp performance drop on ArEnAV.

    Table 10b extends this to modern SOTA detectors (LipForensics, M2TR, LAA-Net (CVPR-24), ForensicsAdaptor (CVPR-25)): they remain strong on FF++, DFDC, CelebDF, etc., but collapse to near-random performance on ArEnAV. This shows that the difficulty is not due to “outdated” models.

- **Metric suitability.**

    Table 4 (added after rebuttal) provides forged-segment duration statistics and shows that fake spans in ArEnAV are short but not vanishingly small (mean ~0.7s, median ~0.63s, ~12% of clip duration, and 2.1 times longer than in AV-1M). This directly addresses **Wf9J’s** concern that AP@IoU=0.5 might simply be too strict for very short segments.


Together, these results indicate that:

(i) fine-tuning on ArEnAV does not fix the problem,

(ii) strong generalization models still fail on ArEnAV specifically, and

(iii) this is not explained by trivial metric issues.

**Qualitative: model behavior and human performance**

- **Figures 4 and 5 (model predictions)** (added after rebuttal)**.**

    Figure 4 visualizes BA-TFD+ predictions on fake clips. Errors are not concentrated at splice boundaries, and the model does not simply lock onto obvious insertion artifacts.

    Figure 5 shows predictions on *real* CSW clips: the model frequently spikes on natural code-switching regions and labels them as fake. This strongly suggests that the model is confused by the multilingual/CSW nature of the signal, not by generation glitches.

- **Human study.**

    The user study (19 participants, 15 native Arabic speakers) shows ~60% accuracy in binary detection and poor localization (AP@0.5) even for humans. Reasons selected by participants rarely mention “unnatural code-switching” or “obvious artifacts”; they mostly report subtle audio/AV cues and general uncertainty. This is consistent with the CSW manipulations being linguistically natural yet very hard to spot.

---

> ### Author Response · Authors · 2025-12-02
> **PART 2/2:**
>
> Overall, these quantitative and qualitative results jointly support our core claim: **CSW and multilinguality are the primary reasons ArEnAV is hard for both models and humans.**
>
> We also clarified to **8ZWD** that what they perceived as “missing intra-dataset evaluation” or “unfair cross-dataset comparisons” was already present in the paper; in the updated version, we make this clearer in the text around Tables 8–10, without changing the experimental setup.
>
> ---
>
> ### 3. Additional analyses on dataset quality, balance, and generation choices
>
> We added small, focused analyses (in the appendix) that address specific reviewer concerns:
>
> - **Identity overlap and disjoint test set.**
>
>     We used InsightFace + DBSCAN to estimate identity overlap and found ~41.7% train/val–test overlap. We then evaluated BA-TFD/BA-TFD+ separately on overlapping and non-overlapping subsets and found almost identical performance (Table 12). We also commit to releasing an identity-disjoint test split.
>
> - **Instruction following and semantic change.**
>
>     We quantified the fraction of “meaning+translation” samples where the LLM barely changed the semantics (entailment > 0.8): this is a small fraction of the data (Table 13). Splitting the test set by this threshold yields almost identical performance for BA-TFD/BA-TFD+, indicating that these few “near-paraphrase” cases do not drive the difficulty.
>
> - **TTS and lip-sync components.**
>
>     An ablation table reports performance across 4 TTS and 2 lip-sync models (Table 14). Lower-quality TTS is indeed easier to detect, whereas higher-fidelity XTTS-based pipelines reduce detectability; Diff2Lip and LatentSync behave similarly for AV models. This shows that higher generation quality makes the benchmark harder, as intended.
>
> - **Class imbalance.**
>
>     We clarify that all training uses subsampling to keep real/fake balanced, so the observed trends are not artifacts of class imbalance.
>
>
> These points directly address several of **8ZWD’s** individual criticisms (identity split, metadata, imbalance, generation details, training protocol in Table 10).
>
> ---
>
> ### 4. Linguistic and human validation of CSW naturalness
>
> We back the linguistic plausibility of our CSW with both automatic and human evidence:
>
> - **Automatic linguistic validation.**
>
>     We compare perplexity and bidirectional entailment of original vs. manipulated transcripts (Figure 3b) using large Arabic–English LMs (e.g., Jais-3B, Qwen-2.5-7B), and against an external CSW text corpus. Real and fake transcripts have very similar perplexity, and the entailment distribution shows meaningful semantic changes without collapsing into trivial paraphrases.
>
> - **Human validation.**
>
>     Native Arabic speakers in the user study report that the clips sound natural and are often indistinguishable from genuine content, even though they find the detection task hard. Very few participants attribute their decisions to “unnatural CSW” (Table 5,6).
>
>
> This addresses **W12a**/**Wf9J’s** request for stronger evidence that our LLM-generated CSW resembles real CSW patterns, rather than being an artificial artifact of prompting.
>
> ---
>
> ### 5. Ethics, privacy, dual-use and reproducibility
>
> Finally, we clarified the ethics and reproducibility aspects:
>
> - We follow the common practice of existing audio-visual datasets (LRS3, VoxCeleb, AVSpeech, FakeAVCeleb, AV-1M) by using only public YouTube content for non-commercial research (Lines 548-551), and we gate access via a strict EULA (lines 552-564) that prohibits re-identification, redistribution, or harmful use and requires institutional affiliation and an IRB application ID.
> - The human study followed university IRB procedures (adult participants, voluntary participation, no personal data); the IRB ID will be included in the camera-ready version (Lines 586-593).
> - We will release the dataset, metadata (including real–fake pairings), prompts, scripts, and manifests so that the community can reproduce our benchmarks even if specific APIs/models change.

---

### Meta-Review · Area_Chair_BP5a · 2026-01-08

**Summary:**

This is a borderline case.

The paper receives 3 high-quality reviews, with 2 negative initial ratings (4, 4) and 1 positive initial rating (8).

No reviewer has participated in the discussions.

One negative reviewer (rating: 4) has main concerns in contribution (without detection model or algorithm) and reproducibility (reliance on API). The AC also agrees with this reviewer, but as the multilingual and code-switching problem has not been systematically studied, a dataset (as well as the creation pipeline) and a benchmarking are acceptable.

Another negative reviewer (rating: 4) has main concerns in lack of experiment and analysis for the multilingual and code-switching problem. The authors have given responses to these comments.

The positive reviewer has also not participated in the discussions and the authors have given detailed responses and additional results.

**Reviewer Concerns:**

See the above summary.

**Reviewer Scores:**

See the above summary.

---

### Decision · Program_Chairs · 2026-01-26

Accept (Poster)